# LoRAGen: Structure-Aware Weight Space Learning for LoRA Generation

**Hao Huang**[1]    **Jingtao Ding**[†2]    **Mengqi Liao**[1]    **Xin Wang**[1]    **Jinyang Ban**[1]
**Yuan Yuan**[3]    **Huaiyu Wan**[1]    **Yong Li**[†2]

[1]**Beijing Jiaotong University**    [2]**Tsinghua University**    [3]**New York University**
haohuang@bjtu.edu.cn, {dingjingtao, liyong07}@tsinghua.edu.cn

## Abstract

The widespread adoption of Low-Rank Adaptation (LoRA) for efficient fine-tuning of large language models has created demand for scalable parameter generation methods that can synthesize adaptation weights directly from task descriptions, avoiding costly task-specific training. We present LoRAGen, a structure-aware method for generating LoRA parameters from natural language descriptions. Through empirical analysis of LoRA libraries, we identify two key structural properties of LoRA parameter spaces: non-uniqueness of low-rank decomposition and heterogeneous weight distributions across network modules. These properties necessitate specialized parameter generation methods rather than general weight space learning approaches. LoRAGen employs a latent diffusion model with two innovations: weight-space supervision on full adaptation matrices to handle decomposition non-uniqueness, and a module-aware Mix-of-Experts decoder that adapts to module-specific weight distributions. Experiments show LoRAGen achieves 96.0% performance relative to task-specific LoRAs on FLAN-T5-large and 72.7% on Gemma-2-2B-Instruct for in-distribution tasks, while obtaining 40.2% on zero-shot generation across unseen tasks—surpassing baselines by nearly 5%. Our work establishes the first structure-aware approach to LoRA generation with insights into adaptation weight space geometry. The implementation of our approach is available: `https://github.com/tsinghua-fib-lab/LoRAGen`.

## 1 Introduction

Exploring the weight space of neural networks, i.e., the high-dimensional parameter space spanned by populations of trained networks, has emerged as a powerful paradigm for understanding model mechanisms and enabling novel applications. Parameter generation, which trains models to produce weights for target networks, represents a particularly promising direction that has gained significant attention in recent years (Schürholt et al., 2021b; Schürholt et al., 2022; Schürholt et al., 2024; Wang et al., 2024). The rise of large language models (LLMs) has created opportunities to apply parameter generation techniques in this domain, particularly through Low-Rank Adaptation (LoRA) (Hu et al., 2022) generation, the direct synthesis of LoRA parameters for efficient fine-tuning. While traditional LoRA workflows require task-specific training with custom datasets and hyperparameters, creating engineering overhead and limiting reusability (He et al., 2022a; Lv et al., 2024), LoRA generation enables direct parameter synthesis from natural language task descriptions, improving scalability and unlocking adaptive model behavior without maintaining extensive adapter libraries.

Recent work generates LoRA parameters conditioned on task identifiers, datasets, or natural language task descriptions. A common design involves learning a hypernetwork—a neural network that generates parameters for another base network (Ha et al., 2016). With advances in deep generative models, one category of approaches learns a lower-dimensional representation directly from the weight space and decodes this latent representation into LoRA parameters (Shao et al., 2025). However, the underlying encoder-decoder model must encode the entire LoRA parameters at once

---

[†]Corresponding author.

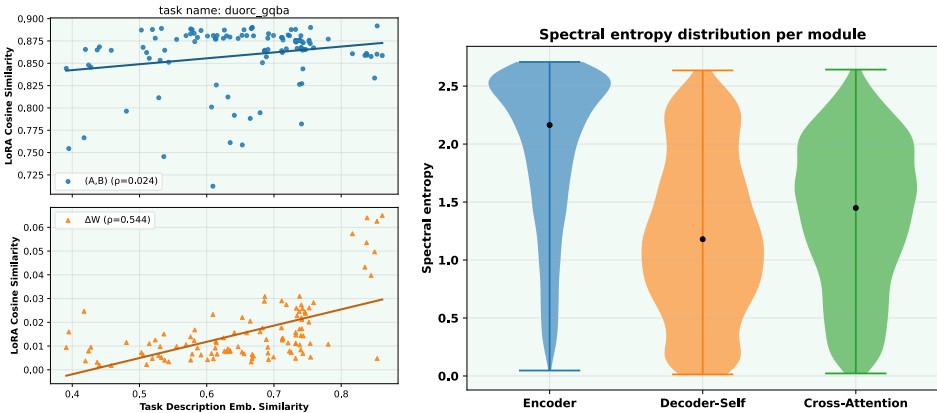

Figure 1: **Two empirical observations.** *Left:* Task description embedding similarity versus LoRA parameter similarity (measured by full adaptation matrix and low-rank decomposition matrix, respectively). *Right:* Spectral-entropy distributions of LoRA parameters for `FLAN-T5-large`, grouped by module type: encoder self-attention, decoder self-attention, and cross-attention.

into the learned latent representation, which limits the size of LoRA that can be embedded. Another category learns a conditional diffusion prior over the latent space to generate LoRA parameters from random noise based on specific task conditions at test time, but these approaches struggle to generate well-performing LoRA parameters across diverse architectures and datasets (Jin et al., 2024; Liao et al., 2024; Wu et al., 2024; Soro et al., 2025). A recent work, Text-to-LoRA (T2L) (Charakorn et al., 2025), employs a hypernetwork trained to construct LoRA parameters in a single inexpensive forward pass, enabling zero-shot generation for entirely unseen tasks based on the hypothesis that different LoRAs share the same adaptation mechanism and can be optimized without explicit structure. However, these methods treat LoRA parameter generation as an instantiation of general weight space learning approaches Schürholt et al. (2024); Wang et al. (2024), while lacking a tailored design specifically for LoRA characteristics.

In this work, we begin with an empirical analysis of a LoRA library built on the Transformer-based model `FLAN-T5-large`(Chung et al., 2024) (Figure 1). Our analysis reveals the **non-uniqueness of low-rank decomposition** as a key challenge. As shown in Figure 1 (left), we observe a clear positive correlation between adapter similarity in the weight space and task description embedding similarity, but we find no correlation between the cosine similarity of low-rank decomposition matrices and task description embedding similarity. This suggests that supervision in the full adaptation matrix space should generalize better than element-wise reconstruction. However, most LoRA generators reconstruct low-rank decomposition matrices directly (Ha et al., 2016; Jin et al., 2024; Liao et al., 2024; Soro et al., 2025; Charakorn et al., 2025; Lv et al., 2024), which can make training sensitive to arbitrary rescalings or rotations of the low-rank decomposition matrices that produce the same full adaptation matrix (Gabrielsson et al., 2024). Additionally, in Figure 1 (right), we identify **significant heterogeneity in LoRA weight distributions across different modules**. The spectral-entropy (Yunis et al., 2024; Roy & Vetterli, 2007) distributions differ systematically by module type within `FLAN-T5-large`: encoder self-attention exhibits higher entropy, decoder self-attention shows the lowest entropy, and cross-attention lies in between, indicating heterogeneous weight distributions across parts of the base model. However, existing methods typically use a single decoder across modules (Lv et al., 2024; Charakorn et al., 2025), which overlooks the module structure (Ostapenko et al., 2024) and can mismatch the spectral-entropy distributions.

Motivated by the two empirical observations that reveal distinct properties of the LoRA weight space, we introduce *LoRAGen*, a method that generates LoRA parameters from natural-language task descriptions for structure-aware LoRA weight space learning. Specifically, we use a LoRA weight autoencoder (LAE) to learn a latent space over LoRA parameters and a conditional latent diffusion model conditioned on natural language task descriptions to generate latent representations from random noise, followed by a decoder that processes these generated representations to produce new LoRA parameters. To address the non-uniqueness issue, we supervise the full adaptation matrix

directly rather than the low-rank decomposition matrices, proposing two weight space loss terms: a direction loss and a spectral loss on the full adaptation matrix. This avoids sensitivity to rescalings or rotations of low-rank decomposition matrices that yield the same full adaptation matrix, leading to more consistent and task-relevant weight space learning. To address the heterogeneity of weight distributions, we introduce a module-aware Mix-of-Experts (MoE) decoder with routing that uses a structural embedding combining a latent variable with learnable module and layer embeddings. This allows experts to specialize to the observed module-specific weight distribution patterns while ensuring a controlled sharing mechanism via the chosen expert pool configuration, facilitating generalization across different architectures of the base model.

Our key contributions are: (1) We propose the first structure-aware weight learning method tailored to the LoRA weight space, enabling effective LoRA parameter generation for diverse downstream tasks. (2) LoRAGen introduces weight-space losses on the full adaptation matrix to address non-uniqueness of low-rank decomposition and employs a module-aware MoE decoder to model heterogeneous LoRA weight distributions. (3) LoRAGen achieves strong in-distribution LoRA generation performance close to task-specific LoRAs across architectures: 96.0% on `FLAN-T5-large`, 72.7% on `Gemma-2-2B-Instruct` and further achieves 40.2% in zero-shot LoRA generation on seven unseen tasks, surpassing competitive baselines by nearly 5%.

## 2 RELATED WORK

**Weight Space Learning for Parameter Generation.** Research on weight space learning has followed two main directions: predicting model properties from trained weights (Schürholt et al., 2021a; Unterthiner et al., 2020; Zhao et al., 2025) and generating new parameters for neural networks (Schürholt et al., 2022; Yuan et al., 2024; Schürholt et al., 2024; Li et al., 2025; Wang et al., 2025; Saragih et al., 2025; Gupta et al., 2026). Here, we focus on parameter generation. With advances in deep generative models, one category of approaches learns a latent representation directly over populations of trained models and decodes this latent representation to generate parameters (Berardi et al., 2022). Another category employs conditional latent diffusion models to generate target parameters (Peebles et al., 2022; Soro et al., 2024). Recent studies introduce graph-based encoders that treat networks as graphs and enable parameter generation across architectures (Kofinas et al., 2024). These approaches have been applied in areas such as meta-learning, transfer learning, and model compression (Finn et al., 2017; Nava et al., 2022; Wang et al., 2023). Overall, prior work exploring parameter generation in weight space (Peebles et al., 2022; Berardi et al., 2022; Schürholt et al., 2024) demonstrates that such methods can generalize across a range of architectures and datasets.

**Hypernetworks for LoRA Generation.** A hypernetwork is a neural network designed to generate the parameters of another "base" network, providing an indirect encoding of the base model's weights (Ha et al., 2016; Zhang et al., 2018; Schug et al., 2025). With the development of deep generative models, one category of approaches (Shao et al., 2025) learns conditional latent representations of pretrained LoRA parameters for new LoRA parameter generation. Another category employs conditional latent diffusion priors over latent space (Wu et al., 2024; Jin et al., 2024; Soro et al., 2024), enabling the generation of task-specific LoRA parameters. Recently, hyperLoRA (Lv et al., 2024) employs instruction-tuned hypernetworks with constrained loss and demo selection to produce stable and generalizable adapters. While these approaches have advanced multi-task LLM adaptation, they typically rely on learned task identifiers, limiting their capacity for zero-shot LoRA parameter generation to unseen tasks (Ivison & Peters, 2022; Mahabadi et al., 2021; He et al., 2022b; Schürholt et al., 2021a; Ortiz-Barajas et al., 2024; Lv et al., 2024). Recent work explores natural language as conditioning signals for zero-shot generation (Xiao et al., 2023; Ivison et al., 2023; Phang et al., 2023), with T2L (Charakorn et al., 2025), DnD (Liang et al., 2025), and LoRA-Gen (Xiao et al., 2025) utilizing hypernetworks to generate LoRA adapters from textual prompts. Futhermore, (Park et al., 2026; Wójcik et al., 2026; Liu et al., 2026; Charakorn et al., 2026) treat LoRA generation as a general weight space learning problem, overlooking the unique structural properties of LoRA parameter spaces. In contrast, LoRAGen is the first weight space learning approach that specifically accounts for the structural characteristics of LoRA spaces, leading to more effective parameter generation.

## 3 PRELIMINARIES

**Low-Rank Adaptation (Hu et al., 2022):** Low-rank matrix $\Delta W$ serves as a adapter to a base model. For a pretrained weight matrix $W_0$, the fine-tuned linear transformation is given by $h = W_0 + \Delta W = W_0 + BA$, where $A \in \mathbb{R}^{r \times d_{\text{in}}}$ and $B \in \mathbb{R}^{d_{\text{out}} \times r}$ are low-rank decomposition matrices with $r \ll \min\{d_{\text{in}}, d_{\text{out}}\}$. We ignore the module type and layer index of the LoRA parameters when referring to all LoRA parameters. Therefore we index them by a module type $m \in \mathcal{M}$ (e.g., query projection) and a layer index $\ell \in \{1, \ldots, L_m\}$. A LoRA adapter at each location $(m, \ell)$ specifies a low–rank matrix $\Delta W_{m,\ell} = B_{m,\ell} A_{m,\ell}$. We denote the low-rank adapter by $\Delta W := \{\Delta W_{m,\ell}\}_{m \in \mathcal{M}, 1 \leq \ell \leq L_m}$. Note that the low–rank decomposition is *not unique*: for any invertible matrix $R \in \mathbb{R}^{r_{m,\ell} \times r_{m,\ell}}$, $(B_{m,\ell}R)(R^{-1}A_{m,\ell})$ produces the same low-rank matrix $\Delta W_{m,\ell}$.

**Problem setting.** We assume an LoRA library of pairs $\mathcal{D} = \{(x^{(i)}, \Delta W^{(i)})\}_{i=1}^N$, where $x^{(i)}$ is a natural language description of task $t^{(i)}$ and $\Delta W^{(i)} = \{\Delta W_{m,\ell}^{(i)}\}$ represents the fine-tuned low-rank adapter for task $t^{(i)}$. Our goal is to train a LoRA generator using $\mathcal{D}$ that produces new LoRA parameters $\widehat{\Delta W}'$ given a natural language task description $x'$, where $x'$ may be either in-distribution (from $\mathcal{D}$) or out-of-distribution (unseen tasks).

## 4 THE PROPOSED METHOD: LORAGEN

### 4.1 DESIGN PRINCIPLES FROM EMPIRICAL OBSERVATIONS

We first summarize two observations (Obs) over a library of low-rank adapters (two panels in Figure 1), which motivates our tailored design in LoRAGen. The details of observations experiment implementation is reported at the Appendix A.1.

**Obs-1: Non-uniqueness of the low-rank decomposition.** Here we focus on the non-uniqueness property of low-rank decompositions in LoRA. Specifically, we examine a LoRA library trained on `FLAN-T5-large`. While $\Delta W$ is uniquely defined as a full adaptation matrix, its low-rank decomposition matrices $(A, B)$ is not unique, illustrated in Sec. 3. This property motivates the following hypothesis: if we supervise the low-rank decomposition matrices $(A, B)$ directly, like element-wise reconstruction, training becomes sensitive to arbitrary rescalings and rotations that still yield the same full adaptation matrix $\Delta W$. By supervising the full adaptation matrix itself, we avoid this ambiguity and directly align the generated adapters with the pretrained adapters in the full adaptation matrix space, which should lead to more consistent and task-relevant weight space learning.

We test this hypothesis by computing the *pairwise* similarity between its LoRA adapter and each of the other 111 adapters in the FLAN subset for a representative task which is selected from the four major task categories in the FLAN subset so that it reflects the overall distribution of the dataset rather than any specific task; further details are reported in Appendix A.1. To measure adapter similarity, we compute the cosine similarity of the concatenation of flattened low-rank A and B matrices of all layers and flattened $\Delta W$, respectively. To avoid scale mismatch between the two similarity measures, they are plotted in separate subplots, each fitted with its own least-squares trend line. We observe a clear positive correlation between the task embedding similarity and the adapter similarity in the weight space, whereas the similarity measured on the low-rank decomposition matrices exhibits near-zero Spearman coefficients $\rho$, indicating the lack of correlation. This phenomenon aligns with the non-uniqueness property of low-rank decompositions in LoRA and suggests that supervision in the full adaptation matrix space should generalize better than element-wise reconstruction.

Motivated by Obs-1, we therefore introduce *adapter-level supervision* (Sec. 4.3), where losses are defined directly in the weight space of $\Delta W$. Concretely, we introduce two weight space loss terms, combining a *direction loss* $\mathcal{L}_{\text{ang}}$ (Eq. 2) that aligns normalized LoRA directions, with a *spectral loss* $\mathcal{L}_{\text{spec}}$ (Eq. 3) that matches leading singular values. These objectives enforce task-consistent LoRA generation while remaining robust to the inherent non-uniqueness of low-rank decompositions.

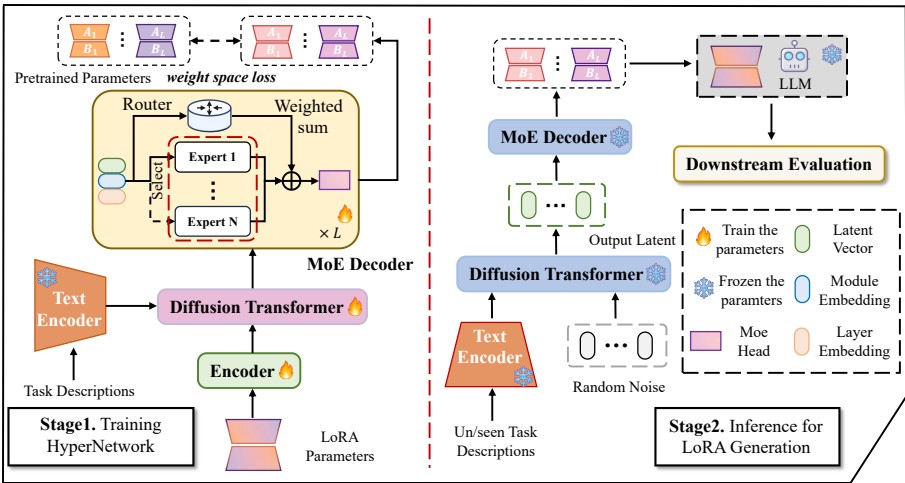

Figure 2: **Overall framework of LoRAGen.** Our approach consists of two stages: First, we train hypernetwork based on LoRA weight autoencoder to encode and reconstruct LoRA parameters, and diffusion process conditioned on natural language task descriptions to predict denoised latent. Second, random noise and un/seen natural language task descriptions are fed into LoRAGen to generate LoRA parameters, which can be incorporated with the LLM to evaluate downstream tasks.

**Obs-2: Heterogeneity of LoRA weight distributions.** Here we focus on the heterogeneity of LoRA weight distributions across different module types in a transformer-based architecture, specifically the `FLAN-T5-large` model, which serves as the base model for LoRA adapters. For each adapter $\Delta W_{m,\ell}$ at a module–layer location $(m, \ell)$, we analyze how its Frobenius energy is distributed across singular directions. Let $\{\sigma_i\}$ denote the singular values of $\Delta W_{m,\ell}$ and define the normalized spectrum $p_i = \sigma_i^2 / \sum_j \sigma_j^2$. We then compute the *spectral entropy* $H_{\text{spec}}(\Delta W_{m,\ell}) = -\sum_i p_i \log p_i$, which quantifies the uniformity of energy over directions: low entropy indicates that the energy concentrates in a few dominant directions, indicating effectively lower rank structure, whereas high entropy corresponds to a more even spread of energy (Roy & Vetterli, 2007; Yunis et al., 2024).

For `FLAN-T5-Large`, we reports the spectral-entropy distributions for three module types in `FLAN-T5-large`: Encoder self-attention, Decode self-attention, and *Cross-Attention*. We observe systematic differences across modules: encoder adapters exhibit higher spectral entropy which means energy spread over more directions, decoder self-attention shows the lowest entropy with more concentrated and effectively lower-rank structure, and cross-attention lies in between. This pattern implies that LoRA weight distributions are not homogeneous across the network which means the way energy is distributed over singular directions differs consistently by module type. More results are reported in Appendix A.3.

Motivated by Obs-2, we employ a *module-aware MoE decoder* (Sec. 4.4)to match these module-specific patterns of energy distributions. Specifically, routing is conditioned on a structural embedding that combines a latent variable with learnable module and layer embeddings. The decoder can operate with either (i) a single shared expert pool for all locations *or* (ii) separate expert pools per module type (e.g., encoder attention). Per-module output heads $\mathcal{H}_m$ map the gated expert outputs into LoRA parameters $\widehat{\Delta W}_{m,\ell}$. This architecture lets experts specialize to the observed module-specific energy distribution patterns (e.g., lower-entropy/low-rank tendencies in Decoder-Self versus higher-entropy Encoder), while the router and the chosen pool configuration ensures controlled sharing mechanism without losing module-specific specialization.

## 4.2 METHOD OVERVIEW

As shown in Figure 2, the overall framework of *LoRAGen* can be divided into two stages:

First, given pretrained LoRA parameters $\Delta W$, the LAE ($\mathcal{E}_\phi, \mathcal{D}_\theta$) encodes $\Delta W$ into per-location latents with a diagonal–Gaussian posterior $q_\phi(z \mid \Delta W)$, and a module-aware MoE decoder $\mathcal{D}_\theta$ (Sec. 4.4) decodes $z$ to full LoRA parameters $\widehat{\Delta W}$. Unlike a standard VAE that uses an reconstruction likelihood, we train LAE using *adapter-level structural supervision* directly in the LoRA weight space (Sec. 4.3), i.e., direction and spectral losses defined on $\Delta W_{m,\ell}$. We then train a diffusion transformer to model the conditional prior $p_\psi(z_0 \mid c)$ over LAE latents, where $c$ denote the embedding of a natural-language task description produced by the text encoder (Figure 2). Specifically, we sample a latent target $z_0 \sim q_\phi(z \mid \Delta W)$, apply a forward noising process $q(z_t \mid z_0) = \mathcal{N}(\sqrt{\bar{\alpha}_t}z_0, (1-\bar{\alpha}_t)I)$, and train a denoiser $f_\psi$ conditioned on $c$ to predict the denoising target:

$$\mathcal{L}_{\text{diff}}(\psi) = \mathbb{E}_{z_0 \sim q_\phi(\cdot \mid \Delta W),\, t,\, \epsilon \sim \mathcal{N}(0,I)}\left[\|v - f_\psi(z_t, t, c)\|_2^2\right], \tag{1}$$

where $z_t = \sqrt{\bar{\alpha}_t}z_0 + \sqrt{1-\bar{\alpha}_t}\epsilon$. Further diffusion details are reported in Appendix A.6.

Second, at inference, we feed the task description embedding $c$ and Gaussian noise $z_T \sim \mathcal{N}(0, I)$ into the diffusion transformer to run the reverse process and obtain a denoised latent $z_0 \sim p_\psi(z_0 \mid c)$. We then pass $z_0$ through the *frozen* MoE decoder $\mathcal{D}_\theta$ from Stage-1 to generate full LoRA parameters for the LLM. Note that the LAE encoder is *only* used during Stage-1 training to obtain latent targets; at inference we sample latents from the diffusion prior and decode them without using the encoder.

## 4.3 Adapter-level Supervision in LoRA Weight Space

Distinct pairs of low-rank decomposition matrices $(A, B)$ produce the same low-rank adapter $\Delta W$. If we choose to generate $A$ and $B$ separately, the LoRA generator must to commit to a specific decomposition of $\Delta W$, even though many different low-rank decomposition matrices produce the identical adapter. To avoid this ambiguity induced by non-uniqueness property of the low-rank decomposition, we directly add supervision signal at the level of low-rank adapter $\widehat{\Delta W}_{m,\ell} = \mathcal{D}_\theta(z)_{m,\ell}$, and introduce two weight space loss terms: (i) a *direction* loss based on cosine similarity that depends on the low-rank adapters after normalizing their Frobenius norm. (ii) a *spectral* loss that matches the leading singular values of the predicted and pretrained low-rank adapters.

**Direction loss.** Because many pairs of low-rank decomposition matrices $(A, B)$ produce the same low-rank adapter $\Delta W$, supervising $A$ and $B$ individually can introduce arbitrary differences in the Frobenius norm of $\Delta W$. To make supervision insensitive to it, we normalize both the predicted and target low-rank adapters to unit Frobenius norm and compare their directions directly to capture the per-task direction. Thus we introduce a *direction* loss that measures the angular mismatch between the predicted and target low-rank adapters:

$$\mathcal{L}_{\text{ang}}(m, \ell) = 1 - \frac{\langle \widehat{\Delta W}_{m,\ell}, \Delta W_{m,\ell} \rangle_F}{\|\widehat{\Delta W}_{m,\ell}\|_F \|\Delta W_{m,\ell}\|_F}. \tag{2}$$

where $\langle \cdot, \cdot \rangle_F$ denotes the Frobenius inner product and $\|\cdot\|_F$ is the Frobenius norm.

**Spectral loss.** Direction supervision does not capture how the squared Frobenius norm is distributed across the *singular spectrum*. Two low-rank adapters can have similar cosine similarity yet differ in their *leading singular values*, i.e., in the proportion of the squared Frobenius norm explained by the top singular directions. To account for this, we introduce a *spectral* loss that matches the leading singular values of the two low-rank adapters:

$$\mathcal{L}_{\text{spec}}(m, \ell) = \left\|\boldsymbol{\sigma}_{1:k_{m,\ell}}(\widehat{\Delta W}_{m,\ell}) - \boldsymbol{\sigma}_{1:k_{m,\ell}}(\Delta W_{m,\ell})\right\|_{p, \boldsymbol{\omega}_{m,\ell}} \tag{3}$$

where (i) $\sigma_i(X)$ denotes the $i$-th singular value of $X$, and $\boldsymbol{\sigma}_{1:k}(X) := (\sigma_1(X), \ldots, \sigma_k(X))$ lists the top-$k$ values in nonincreasing order; (ii) $r_{m,\ell}$ is the LoRA rank at location $(m, \ell)$, and $k_{m,\ell} \in \{1, \ldots, r_{m,\ell}\}$ is the smallest integer such that the top-$k$ singular values of the target low-rank adapter explain at least a fraction $\rho \in (0, 1)$ of its squared Frobenius norm. (iii) $\|u\|_{p,\boldsymbol{\omega}}$ is a weighted $\ell_p$ norm with $p \in \{1, 2\}$; (iv) $\boldsymbol{\omega}_{m,\ell} = (\omega_{m,\ell,1}, \ldots, \omega_{m,\ell,k_{m,\ell}})$ are nonnegative normalized singular-value weights.

Thus, we aggregate these two terms together as follows:

$$\mathcal{L}_{\text{adapter}}(\theta, \phi) = \mathbb{E}_{z \sim q_\phi(z|\Delta W)} \left[ \sum_{m \in \mathcal{M}} \sum_{\ell=1}^{L_m} \lambda_{m,\ell} \Big( \alpha_{\text{ang}} \mathcal{L}_{\text{ang}}(m, \ell) + \alpha_{\text{spec}} \mathcal{L}_{\text{spec}}(m, \ell) \Big) \right], \quad (4)$$

where $q_\phi(z \mid \Delta W)$ is the encoder posterior; $\alpha_{\text{ang}}, \alpha_{\text{spec}} > 0$ are hyperparameters balancing the two loss terms; and $\lambda_{m,\ell} \geq 0$ are location weights. Note that $\mathcal{L}_{\text{ang}}$ aligns direction, while $\mathcal{L}_{\text{spec}}$ aligns the leading spectrum (i.e., the proportion of the squared Frobenius norm explained by the top singular directions). Thus $\mathcal{L}_{\text{adapter}}$ aligns the performance of the specific-task low-rank adapter yet remains robust to low-rank decompositions.

The overall objective of training LAE is

$$\mathcal{L}_{\text{LAE}}(\theta, \phi) = \alpha_{\text{adapter}} \mathcal{L}_{\text{adapter}}(\theta, \phi) + \beta \, D_{\text{KL}}\big(q_\phi(z \mid \Delta W) \,\|\, \mathcal{N}(0, I)\big) + \lambda_{\text{moe}} \mathcal{L}_{\text{moe}}(\theta), \quad (5)$$

where $\alpha_{\text{adapter}}, \beta, \lambda_{\text{moe}} > 0$ are scalar coefficients; $\mathcal{L}_{\text{moe}}$ is the MoE load-balancing auxiliary loss(Sec. 4.4).

## 4.4 MODULE-AWARE MOE DECODER

To capture structural heterogeneity across the module types and layers, we further introduce a module-aware MoE decoder $\mathcal{D}_\theta$ that can use either a single shared expert pool for all locations or separate pools per module type (e.g., encoder attention). The notation below treats $E$ as the number of experts in the active pool used by the current location.

**Inputs and routing.** For each location $(m, \ell)$, we form a structural embedding $h_{m,\ell} = \left[ z_{m,\ell}; e_m; e_\ell \right] \in \mathbb{R}^{d_h}$, where $z_{m,\ell} \in \mathbb{R}^{d_z}$ is the latent variable, $e_m \in \mathbb{R}^{d_m}$ and $e_\ell \in \mathbb{R}^{d_\ell}$ are learnable module and layer embeddings. A router with parameters $W_{\text{r}} \in \mathbb{R}^{E \times d_h}$ outputs logits $\ell_{m,\ell} = W_{\text{r}} h_{m,\ell} \in \mathbb{R}^E$ and applies top-$K$ gating:

$$g_{(m,\ell),e} = \frac{\exp(\ell_{m,\ell,e}/\tau)}{\sum_{e' \in S_{m,\ell}} \exp(\ell_{m,\ell,e'}/\tau)} \, \mathbb{I}[\, e \in S_{m,\ell} \,], \quad (6)$$

where $\tau > 0$ is the temperature, $S_{m,\ell} \subset \{1, \ldots, E\}$ is the index set of the top-$K$ experts by logit value, and $\mathbb{I}[\cdot]$ is the indicator function (equal to 1 if its argument is true and 0 otherwise).

**Experts and per-module heads.** Each expert $\mathcal{E}_e$ is a small MLP. The gated output feeds a per-module head $\mathcal{H}_m$:

$$\widehat{\Delta W}_{m,\ell} = \mathcal{H}_m \Big( \sum_{e \in S_{m,\ell}} g_{(m,\ell),e} \, \mathcal{E}_e(h_{m,\ell}) \Big), \quad (7)$$

Note that $\mathcal{H}_m$ is a linear map into a vector followed by a reshape to the full LoRA parameters $\widehat{\Delta W}_{m,\ell} \in \mathbb{R}^{d_{\text{out}}(m,\ell) \times d_{\text{in}}(m,\ell)}$. And its parameters are shared across all layers $\ell$ for the same module $m$.

**Load-balancing auxiliary loss.** To discourage expert collapse we introduce a load-balancing auxiliary loss:

$$\mathcal{L}_{\text{moe}} = \max \Big( E \sum_{e=1}^{E} \bar{p}_e \, \bar{f}_e - 1, \, 0 \Big), \quad (8)$$

where $\bar{p}_e$ is the average gating probability, and $\bar{f}_e$ is the expected fractional load under top-$K$ routing. The details implementation of $\mathcal{L}_{\text{moe}}$ is reported at the Appendix A.11.

## 5 EXPERIMENTS

### 5.1 EXPERIMENTAL SETUP

**Dataset.** In our main experiments, we consider two settings. First, we employ `FLAN-T5-large` (Chung et al., 2024), as the base model. We utilize a subset of FLAN following (Lv et al., 2024)

| Method | AP-Neg | AP-Rec | AP-Pos | QASC-1 | QASC-2 | WQA-T | WQA-A | Avg. (acc) |
|---|---|---|---|---|---|---|---|---|
| `FLAN-T5-Large` | 49.4 | 70.7 | 34.8 | 23.3 | 8.9 | 8.5 | 62.0 | 36.8 |
| Average LoRA | 96.8 | 96.5 | 96.9 | 99.4 | 87.3 | 96.7 | 97.0 | 95.8 |
| D2NWG | 59.5 | 87.3 | 65.9 | 47.0 | 31.1 | 33.4 | 84.8 | 58.4 |
| T2L | 90.5 | 94.1 | 92.7 | 87.9 | 76.8 | 85.5 | 93.3 | 88.7 |
| **Ours** | 96.8 | 96.6 | 97.1 | 99.5 | 87.3 | **97.1** | 97.3 | 96.0 |
| Task-specific LoRAs | 97.2 | 97.0 | 97.8 | 99.7 | 87.3 | 97.0 | 97.3 | 96.2 |

Table 1: Benchmark performance of LoRAGen on FLAN subset (`FLAN-T5-Large` backbone). **Bold numbers** are used when the performance is higher than the task-specific LoRAs.

| Method | ArcC | ArcE | BQ | GSM8K | HS | OQA | PIQA | WG | Avg. (acc) |
|---|---|---|---|---|---|---|---|---|---|
| `Gemma-2-2B-Instruct` | 74.0 | 89.9 | 81.0 | 55.9 | 55.1 | 71.2 | 71.2 | 51.8 | 68.8 |
| Average LoRA | 75.6 | 90.1 | 81.4 | 56.1 | 56.5 | 73.8 | 72.8 | 53.9 | 70.0 |
| D2NWG | 74.1 | 90.0 | 81.2 | 56.0 | 55.1 | 71.3 | 71.3 | 52.0 | 68.9 |
| T2L | 74.3 | 90.2 | 81.2 | 55.9 | 55.2 | 71.4 | 71.5 | 53.8 | 69.2 |
| **Ours** | 76.6 | **90.7** | 84.1 | **56.4** | 64.1 | 80.2 | 75.0 | 54.2 | 72.7 |
| Task-specific LoRAs | 76.7 | 90.6 | 84.7 | 55.9 | 75.4 | 80.2 | 78.0 | 54.6 | 74.5 |

Table 2: Benchmark performance of LoRAGen on 8 benchmark tasks (`Gemma-2-2B-Instruct` backbone). **Bold numbers** are used when the performance is higher than the task-specific LoRAs.

for training and evaluation. Second, we use `Gemma-2-2b-Instruct` (Team et al., 2024) as the base model and evaluate on 8 widely used benchmark tasks, including Arc-challenge (ArcC) and Arc-easy (ArcE) (Clark et al., 2018), BoolQ (Clark et al., 2019), GSM8K (Cobbe et al., 2021), Hellaswag (HS) (Zellers et al., 2019), OpenBookQA (OQA) (Mihaylov et al., 2018), PIQA (Bisk et al., 2020), and Winogrande (WG) (Sakaguchi et al., 2021). More details about the datasets are reported in the Appendix. For both settings, we extract task embeddings from natural language task descriptions using the `FLAN-T5-large` encoder. Task descriptions for each dataset are fully generated by LLM, as described in the Appendix A.10. For each dataset, We sample and report the average performance of 3 set of LoRA weights sampled with LoRAGen.

**Baseline Setup.** As baselines, we consider task-specific LoRAs, weight-averaged LoRA (Wortsman et al., 2022). We compare D2NWG (Soro et al., 2024), which is a latent diffusion conditioned on datasets for LoRAs generation. We also implement T2L (Charakorn et al., 2025), which is a hypernetwork that generates LoRAs based on natural language task embedding. Reproduction details are provided in the Appendix A.11.

## 5.2 PERFORMANCE COMPARISON

**In-distribution LoRA Generation Performance.** First, we focus on whether LoRAGen can recover the performance of trained LoRAs (Charakorn et al., 2025; Brüel-Gabrielsson et al., 2024), which enables low-rank adaptation with minimal compute requirements. Table 1 reports results on seven FLAN tasks using natural–language task embeddings from the `FLAN-T5-Large` encoder. LoRAGen closely matches the oracle adapters and outperforms D2NWG and T2L on average performance . We think that the gain comes from the design of MoE decoder to capture the heterogeneity of weight distributions across different module types and layers within the structure-aware LoRA weight space.

Moreover, we train a separate LoRAGen model for a decoder-only base model (`Gemma-2-2B-Instruct`) to assess whether the same structure-aware design generalizes across architectures. As shown in Table 2, LoRAGen remains competitive or superior to several baselines and is close to task-specific LoRAs on average performance, indicating that our approach scales from T5-based adapters to decoder-only adapters. In several tasks, such as ArcE, GSM8K and OQA, our method even matches or surpasses task-specific adapters, suggesting that adapter-level supervision captures task-relevant structure rather than memorizing particular LoRA parameters.

| Method | AP-Rec | AP-Pos | QASC-1 | QASC-2 | WQA-T | WQA-A | Avg. (acc) |
|---|---|---|---|---|---|---|---|
| `FLAN-T5-Large` | 70.7 | 34.8 | 23.3 | 8.9 | 8.5 | 62.0 | 34.7 |
| D2NWG | 71.8 | 34.4 | 23.7 | 9.1 | 8.3 | 62.4 | 35.0 |
| T2L | 71.6 | 34.6 | 23.3 | 11.1 | 8.5 | 62.0 | 35.2 |
| **Ours** | **75.1** | **42.2** | **28.1** | **14.5** | **14.3** | **67.2** | **40.2** |

Table 3: Zero-shot generation performance of LoRAGen trained on the **FLAN subset** with `FLAN-T5-Large` as the base model. **Bold numbers** are used to represent the best performance.

| $\mathcal{L}_{\mathrm{ang}}$ | $\mathcal{L}_{\mathrm{spec}}$ | $\mathcal{D}_{\theta}$ | AP-Neg | AP-Rec | AP-Pos | QASC-1 | QASC-2 | WQA-T | WQA-A | Avg. (acc) |
|---|---|---|---|---|---|---|---|---|---|---|
| ✓ | ✓ | ✗ | 59.5 | 87.3 | 65.9 | 47.1 | 31.2 | 33.4 | 84.9 | 58.4 |
| ✗ | ✗ | ✓ | 96.8 | 95.5 | 97.0 | 98.2 | 86.3 | 95.6 | 96.7 | 95.2 |
| ✗ | ✓ | ✓ | 49.5 | 70.9 | 34.4 | 23.3 | 9.1 | 8.4 | 62.7 | 36.9 |
| ✓ | ✓ | ✓ | **96.8** | **96.6** | **97.1** | **99.5** | **87.3** | **97.1** | **97.3** | **96.0** |

Table 4: Ablation study on FLAN subset. Checkmarks indicate enabled components: direction loss $\mathcal{L}_{\mathrm{ang}}$, spectral loss $\mathcal{L}_{\mathrm{spec}}$, and module-aware MoE decoder $\mathcal{D}_{\theta}$. **Bold numbers** are used to represent the best performance.

**LoRA Generation for unseen tasks.** Furthermore, we explore whether LoRAGen can generate LoRA parameters for unseen tasks. We train LoRAGen on 136 tasks of FLAN subset, each with 20 task descriptions. For each task we sample three sets of LoRA weights and report the average accuracy. As shown in Table 3, LoRAGen achieves the best average accuracy 40.2, outperforming D2NWG and T2L by **+5.2** and **+5.0** points, respectively. We observe that D2NWG and T2L reconstructs pre-trained adapters and struggles to generalize to unseen tasks. This phenomenon is align with non-uniqueness of the low-rank decomposition, which indicates that if we supervise the low-rank decomposition matrices directly, training becomes sensitive to arbitrary rescalings and rotations that still yield the same full adaptation matrix, trending to memorize task-specific LoRA parameters. Instead of this, our method supervise the full adaptation matrix directly to avoid this ambiguity and focus on learn task-relevant weight space learning, resulting in better performance to unseen tasks. Details on computational cost and efficiency are reported in the Appendix A.4.

## 5.3 Ablation Study

In this section, we ablate the adapter-level supervision and the module-aware MoE decoder on the seven FLAN tasks (Table 4). Training with the two adapter-level losses $\mathcal{L}_{\mathrm{ang}}$ and $\mathcal{L}_{\mathrm{spec}}$ but without the decoder $\mathcal{D}_{\theta}$ achieves an average accuracy of 58.4. In contrast, enabling the decoder while removing both adapter-level losses and using only the reconstruction loss results in a significant improvement to an average accuracy of 95.2. This improvement is consistent with **Obs. 2**: the module-aware routing and per-module heads in $\mathcal{D}_{\theta}$ effectively capture the heterogeneity in weight distributions across modules and layers, which substantially improves downstream performance.

A counter-intuitive result occurs when $\mathcal{D}_{\theta}$ is combined with the *spectral* loss but the *direction* loss is omitted: the average accuracy drops to 36.9. In this case, $\mathcal{L}_{\mathrm{spec}}$ only enforces alignment of the magnitudes of the top-$k$ singular values, without constraining the corresponding directions of the left and right singular vectors. As a result, the decoder can match the singular-value magnitudes correctly, but assign them to the wrong directions, which leads to an incorrect weight-space learning and consequently poor task performance. In the zero-shot setting reported in Appendix A.2, Table 5, the same combination yields a much larger gain of +3.6 points in average accuracy (from 36.6 to 40.2), again with the largest improvements on QASC-1, QASC-2, and WQA-T, confirming that adapter-level supervision is particularly important for generalization to unseen tasks.

## 5.4 Detailed Analysis

**Hyperparameter Analysis.** We first analyze the effect of the spectral–energy threshold $\rho$ in the spectral loss as shown in Figure 3(a). We conduct this experiment under the same setting as Table 1 Across all tasks, accuracies remain stable as $\rho$ varies from $0.80$ to $1.00$, with only minor fluctuations.

This shows that the adapter-level supervision is robust to the choice of $\rho$, since the leading singular values already capture sufficient spectral information. In practice, we set $\rho = 0.85$ as it provides a better performance while maintaining stability.

We then examine the hyperparameters of the MoE decoder as shown in Figure 3(b). Here we compare shared vs. unshared expert pools, different top-$K$ routing choices, and the number of experts. The results indicate that unshared pools consistently outperform shared ones, and increasing the number of active experts (top-2 vs. top-1) further improves performance, especially on challenging multi-choice and QA tasks. The best configuration is unshared, top-1, $E = 4$, which strikes a balance between accuracy and computational efficiency.

**Structural Embedding Analysis.** In this section, we assess the contribution of the structural embedding in the MoE decoder as shown in Figure 3(c). We compare three variants: (i) without structural embedding (routing only on latent variables), (ii) with structural embedding, and (iii) an oracle trained with task-specific adapters. The results show that removing structural embeddings substantially reduces accuracy, particularly on tasks requiring fine-grained reasoning such as QASC-1/2. Adding structural embeddings closes most of the gap to the oracle, confirming that encoding module-specific latent is critical for capturing the heterogeneous LoRA weight distributions.

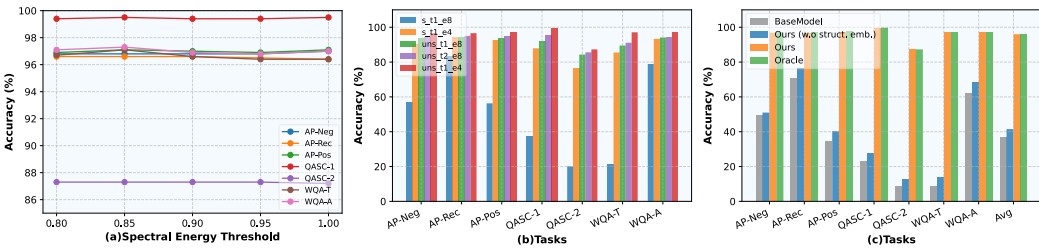

Figure 3: Performance of LoRAGen with different hyperparameters and removing structural embedding. Here, for the middle figure, **s / uns** denote *shared* or *unshared* MoE experts; **t1 / t2** denote *top-1* or *top-2* routing strategies; **e4 / e8** denote using *4* or *8* experts.

## 6  DISCUSSION AND CONCLUSION

We presented *LoRAGen*, a structure-aware approach to LoRA parameter generation grounded in two empirical observations of the LoRA weight space: non-uniqueness of low-rank decompositions and module-wise heterogeneity of weight distributions. Motivated by these observations, we supervise the full adaptation matrix using adapter-level direction and spectral losses, and decode with a module-aware MoE whose routing leverages structural embeddings with shared or per-module expert pools. Empirically, LoRAGen produces strong in-distribution adapters across architectures, closely matching task-specific adapters on `FLAN-T5-large` and `Gemma-2-2B-Instruct` and attains competitive zero-shot performance on unseen tasks from a large LoRA library. Ablation studies show both adapter-level supervision and module-aware decoding are necessary, and sensitivity studies indicate robustness to the spectral-energy threshold and gains from unshared pools with top-$K$ routing. Our results suggest that modeling adapters in the $\Delta W$ space with structure-aware supervision is a promising direction for parameter generation. Nevertheless, LoRAGen still exhibits a performance gap on unseen tasks, indicating that the current conditioning signal and prior may not fully capture the distribution shift implied by novel task descriptions. As future work, we will close this performance gap by improving the task-description encoder and incorporating richer supervision to better align language descriptions with adapte structure and exploring the pool of training adapters and task families to effectively cover a wider task manifold. In addition, diffusion provides a flexible conditional prior but can be computationally expensive when many denoising steps are required. We therefore plan to investigate whether rectified-flow or consistency-based priors can further reduce sampling time for large-scale adapter generation, while retaining the diversity and controllability benefits of LoRA generation.

## ACKNOWLEDGMENTS

This work was supported by the National Natural Science Foundation of China under U24B20180.

## ETHICS STATEMENT

The authors have read and adhered to the ICLR Code of Ethics. The research presented in this paper focuses on the generation of LoRA parameters, with primary applications in LLM domains . All data used for training and evaluation is from publicly available, non-personal scientific datasets, ensuring no privacy concerns. This work does not involve human subjects, and we do not foresee any direct negative societal impacts or risks of perpetuating social biases. Our aim is to advance the development of domain-specific LLM applications.

## REPRODUCIBILITY STATEMENT

The code associated with this paper is available at: `https://github.com/tsinghua-fib-lab/LoRAGen`. It includes the necessary environment configurations and execution scripts. All datasets utilized in this work are publicly accessible. The task descriptions generated from LLM is provided in the Appendix.

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

# A APPENDIX

## A.1 DETAILS FOR OBS-1

As shown in Figure 4, we provide more 9 representative tasks to compare with the remaining 111 tasks on the FLAN subset. These representative tasks are selected so that they cover the three major task categories present in the FLAN subset: Natural Language Understanding (NLU, 98 tasks), Natural Language Generation (NLG, 9 tasks), and Knowledge & Information Extraction (5 tasks). Specifically, we include five NLU tasks (`wikiqa_tpqap`, `wikiqa_jeop`, `fix_punct`, `true_case`, `word_segment`), three NLG tasks (`duorc_tgen`, `duorc_gqba`, `gem_e2e_nlg`), and two Knowledge & Information Extraction tasks (`wiki_bio_what`, `wiki_bio_comp`). This ensures that the visualization reflects the diversity of the dataset rather than any specific task.

Concretely, for a representative task $r$ and another task $t$, we define the *task similarity* as the cosine similarity between their text-description embeddings $e(r)$ and $e(t)$:

$$s_{\text{task}}(r,t) = \frac{e(r)^\top e(t)}{\|e(r)\|_2 \|e(t)\|_2}. \tag{9}$$

For adapter similarity, we consider two variants. Let $\Delta W_r$ and $\Delta W_t$ be the full adaptation matrices for tasks $r$ and $t$, and let $A_r, B_r$ and $A_t, B_t$ be their corresponding low-rank factors. We compute

$$s_{\Delta W}(r,t) = \frac{\langle \text{vec}(\Delta W_r),\ \text{vec}(\Delta W_t)\rangle}{\|\text{vec}(\Delta W_r)\|_2 \|\text{vec}(\Delta W_t)\|_2}, \tag{10}$$

and

$$s_{(A,B)}(r,t) = \frac{\langle v_r,\ v_t\rangle}{\|v_r\|_2 \|v_t\|_2}, \quad v_t = \begin{bmatrix} \text{vec}(A_t) \\ \text{vec}(B_t) \end{bmatrix}. \tag{11}$$

We then report the Spearman correlation coefficient $\rho$ between $\{s_{\text{task}}(r,t)\}$ and each of $\{s_{\Delta W}(r,t)\}$ and $\{s_{(A,B)}(r,t)\}$ over all $t$.

We found that the full adaptation matrix similarity shows a consistent positive correlation with task description embedding similarity. This suggests that tasks with more semantically similar descriptions tend to produce more similar LoRAs, highlighting the presence of structure in the LoRA weight space. This observation also emphasizes the utility of the full adaptation space, as it allows us to capture and exploit underlying similarities between tasks that may not be immediately apparent from the low-rank decomposition matrices. By designing weight space loss based on this similarity, we can better adapt models to generate new LoRA parameters. We note that the absolute cosine similarity between two concatenated full adapters can be close to zero due to the high dimensionality of the concatenated vector; therefore, Figure 4 should be interpreted in terms of correlation rather than magnitude.

## A.2 ABLATION STUDY ON ZERO-SHOT GENERATION

We further examine how the adapter-level losses affect zero-shot generalization on the seven FLAN tasks that are used in Table 1. As summarized in Table 5, enabling only the MoE decoder with a reconstruction loss (i.e., using $\mathcal{D}_\theta$ without $\mathcal{L}_{\text{ang}}$ or $\mathcal{L}_{\text{spec}}$) achieves an average zero-shot accuracy of 36.6. Adding both direction and spectral losses on top of the decoder increases the average accuracy to 40.2 (+3.6 points), with particularly gains on QASC-1 (+3.6), QASC-2 (+4.1), and WQA-T (+4.1). These tasks are among the most challenging reasoning benchmarks in our evaluation, so the improvements provide direct evidence that adapter-level supervision is especially beneficial for zero-shot generalization.

## A.3 WEIGHT DISTRIBUTION ANALYSIS

To further investigate the effect of the module-aware MoE decoder, we analyze the weight distributions of task-specific LoRAs and the adapters generated by LoRAGen on the same evaluation tasks as in the main tables. For the encoder–decoder base model `FLAN-T5-Large`, we use the seven FLAN tasks reported in Table 1. For the decoder-only base model `Gemma-2-2B-Instruct`, we use the eight benchmark tasks reported in Table 2. In each case, we compare three variants: (i) task-specific (oracle) LoRAs, (ii) LoRAGen (full model), and (iii) LoRAGen without the MoE decoder (a single shared decoder).

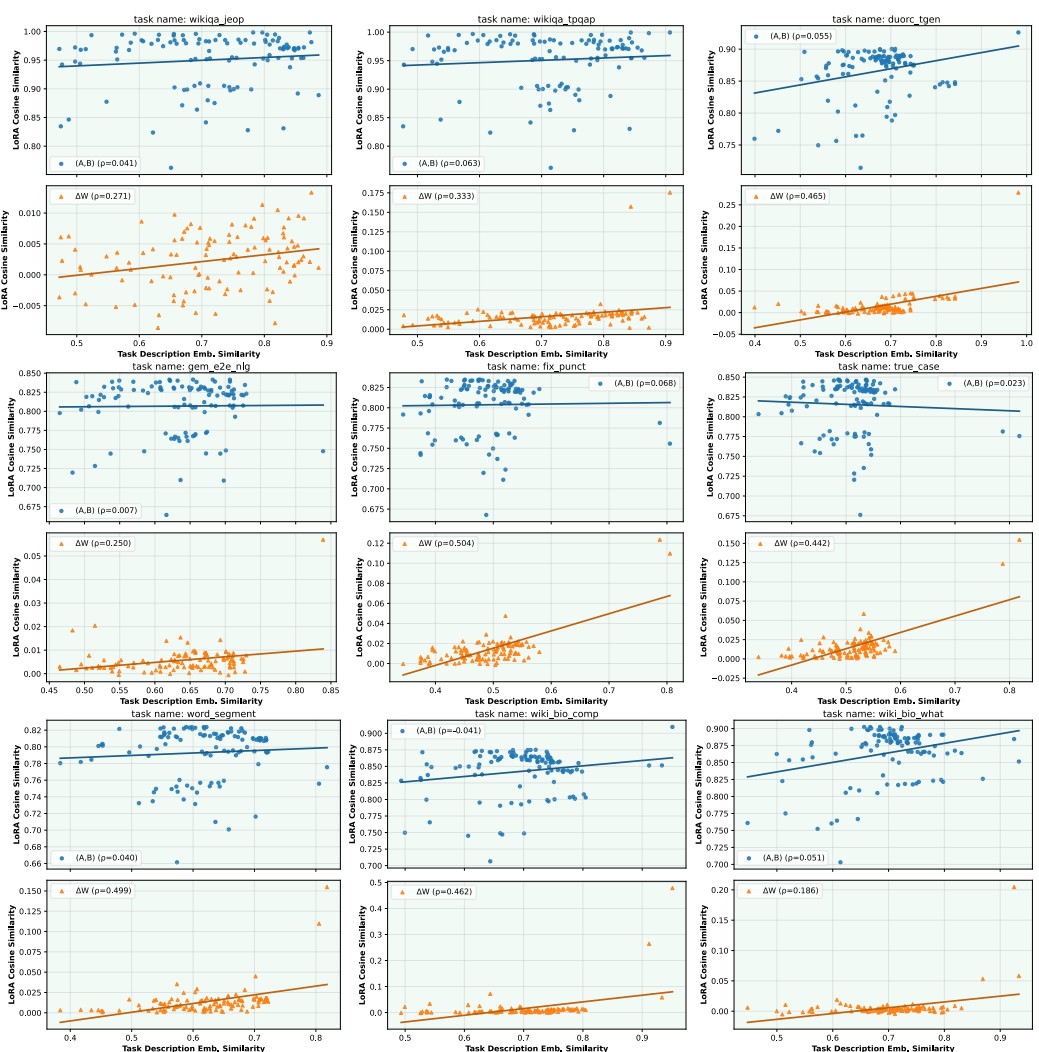

Figure 4: **Relation between LoRA similarity and task description embedding similarity.** Each panel shows the similarity between a representative task adapter and the other 111 adapters trained on a FLAN subset *y***-axis** in the weight space against their similarity in the task embeddings space *x***-axis**. LoRA cosine similarity is measured in two ways: (i) computing cosine on the low-rank decomposition matrices $A$ and $B$ separately (blue dots); (ii) computing cosine on the full adaptation matrix $\Delta W = AB$ (orange triangles). Legends report the Spearman correlation coefficient $\rho$.

| $\mathcal{L}_{\text{ang}}$ | $\mathcal{L}_{\text{spec}}$ | $\mathcal{D}_{\theta}$ | AP-Rec | AP-Pos | QASC-1 | QASC-2 | WQA-T | WQA-A | Avg. (acc) |
|---|---|---|---|---|---|---|---|---|---|
| ✗ | ✗ | ✓ | 72.7 | 35.9 | 24.5 | 10.4 | 10.2 | 65.9 | 36.6 |
| ✓ | ✓ | ✓ | **75.1** | **42.2** | **28.1** | **14.5** | **14.3** | **67.2** | **40.2** |

Table 5: Zero-shot ablation study on the FLAN subset. Checkmarks indicate enabled components: direction loss $\mathcal{L}_{\text{ang}}$, spectral loss $\mathcal{L}_{\text{spec}}$, and module-aware MoE decoder $\mathcal{D}_{\theta}$. **Bold numbers** represent the best performance.

**Spectral entropy distribution analysis.** For each adapter $\Delta W_{m,\ell}$ we compute its spectral entropy $H_{\text{spec}}(\Delta W_{m,\ell})$. For `FLAN-T5-Large`, Figure 5 shows the distributions of $H_{\text{spec}}$ grouped by module type (Encoder, Decoder-Self, Cross-Attention) for the three variants above. Task-specific adapters exhibit clear heterogeneity: encoder modules have the highest spectral entropy, decoder self-attention the lowest, and cross-attention lies in between. The adapters generated by the full

LoRAGen model faithfully capture this pattern, indicating that the MoE decoder has learned module-specific spectral profiles. In contrast, removing the MoE decoder collapses the three module types into almost identical high-entropy distributions, suggesting that a single shared decoder fails to capture the heterogeneous energy patterns across modules.

For `Gemma-2-2B-Instruct`, Figure 6 plots the spectral entropy distributions of decoder adapters for oracle LoRAs, LoRAGen, and LoRAGen without MoE. Task-specific LoRAs concentrate around relatively high entropy values, and the full LoRAGen model shows a very similar, slightly higher-entropy profile. In contrast, removing the MoE decoder shifts towards lower entropy and yields a much broader distribution, indicating that the shared decoder fails to match the spectral distribution of task-specific LoRAs.

**Layer-wise cosine similarity analysis.** Beyond spectral statistics, we also study the alignment between generated and task-specific adapters. Figures 7 and 8 report, for each layer and module type, the average cosine similarity between the generated adapter and the corresponding task-specific adapter, averaged over the seven FLAN tasks in Table 1 and the eight benchmark tasks in Table 2, respectively. With the full LoRAGen model, cosine similarities are consistently positive for `FLAN-T5-Large` and `Gemma-2-2B-Instruct`, indicating that the experts capture meaningful, layer-specific adaptation patterns. In contrast, the model without the MoE decoder yields similarities that remain close to zero across almost all layers, showing that it fails to recover the fine-grained per-layer structure even though it can still perform reasonable reconstruction on in-distribution tasks. Thus, the spectral-entropy and cosine-similarity analyses confirm that the MoE decoder learns to specialize to module types and their characteristic spectral patterns.

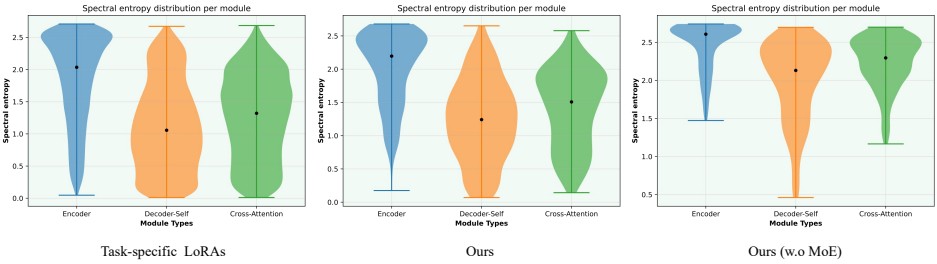

Figure 5: Spectral entropy distributions on `FLAN-T5-Large`. Each group of violins corresponds to Encoder, Decoder-Self, and Cross-Attention modules, respectively, for task-specific LoRAs (left), LoRAGen (middle), and LoRAGen without the MoE decoder (right).

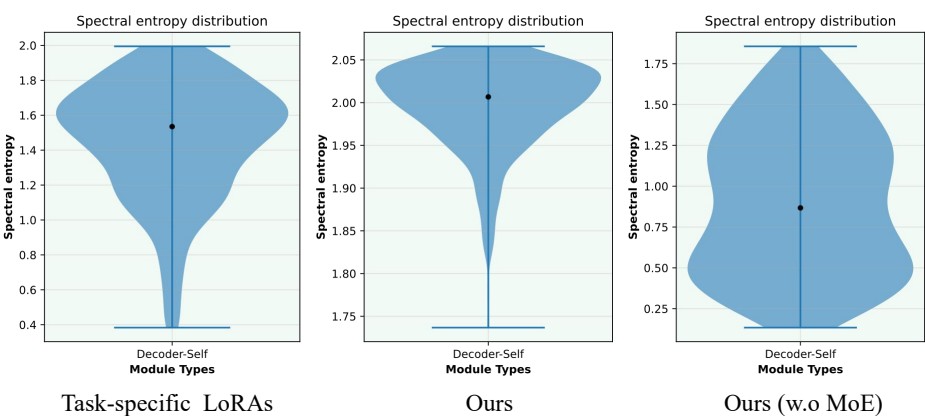

Figure 6: Spectral entropy distributions on `Gemma-2-2B-Instruct` for Decoder-Self module. Task-specific LoRAs (left), LoRAGen (middle), and LoRAGen without the MoE decoder (right).

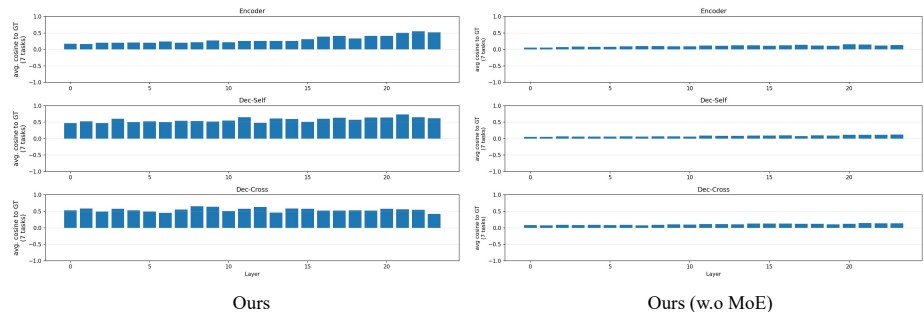

Figure 7: Layer-wise cosine similarity between generated and task-specific LoRA adapters on the FLAN subset with `FLAN-T5-Large` as the base model. For each module type (rows) and layer index (x-axis), we plot the average cosine similarity over 7 evaluation tasks. Left: LoRAGen (full model); right: LoRAGen without the MoE decoder.

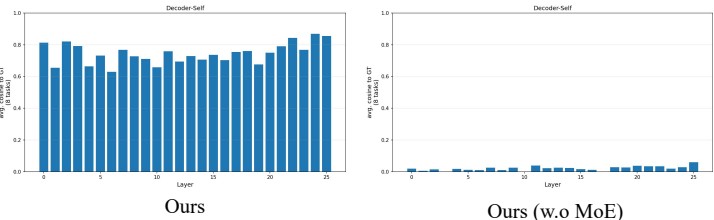

Figure 8: Layer-wise cosine similarity between generated and task-specific LoRA adapters on 8 benchmark tasks with `Gemma-2-2B-Instruct` as the base model. Left: LoRAGen (full model); right: LoRAGen without the MoE decoder.

### A.4 EFFICIENCY OF ADAPTER-LEVEL SUPERVISION

A natural concern is that supervising adapters directly (i.e. losses defined on $\Delta W = AB^\top \in \mathbb{R}^{d \times d}$) could be more expensive than reconstruction on $A, B \in \mathbb{R}^{d \times r}$. We clarify that our implementation avoids any $\mathcal{O}(d^2)$ complexity by using a quadratic form, and thus remains comparable in cost to reconstruction loss.

**Direction loss.** The cosine similarity between two adapters can be computed efficiently as a quadratic form:

$$\langle \Delta W_1, \Delta W_2 \rangle_F = \text{tr}(A_1 B_1^\top B_2 A_2^\top) = \langle A_1, (B_1^\top B_2) A_2^\top \rangle_F,$$

with $\|\Delta W\|_F^2 = \text{tr}(A^\top A B^\top B)$. This requires only $d \times r$ and $r \times r$ multiplications, yielding complexity $\mathcal{O}(dr^2)$ instead of $\mathcal{O}(d^2)$. Hence the direction loss is no more expensive than element-wise reconstruction.

**Spectral loss.** For the spectral loss, we never compute an SVD of the full $d \times d$ adapter. Instead, we compute a reduced QR decomposition. This reduces the problem to an $r \times r$ core matrix $K = R_A R_B^\top$, on which we perform SVD. The resulting complexity is $\mathcal{O}(dr^2 + r^3)$, avoiding any $\mathcal{O}(d^2)$ cost.

**Complexity comparison.** Table 6 reports the per-layer complexities. Both direction and spectral losses are implemented in quadratic or QR decomposition form, avoiding explicit $\mathcal{O}(d^2)$ cost. The overhead relative to element-wise reconstruction loss is bounded by a factor of $r$, which is small in practice.

| Loss term | Per-layer complexity |
|---|---|
| $A$, $B$ reconstruction | $\mathcal{O}(dr)$ |
| Direction (quadratic) | $\mathcal{O}(dr^2)$ |
| Spectral (QR+SVD core) | $\mathcal{O}(dr^2 + r^3)$ |

Table 6: Per-layer complexity of different adapter-level supervision terms. Both direction and spectral losses avoid explicit $\mathcal{O}(d^2)$ cost.

Table 7: Stage 1: LoRA autoencoder configuration for FLAN-T5-Large.

| Item | Configuration |
|---|---|
| Backbone | FLAN-T5-Large (encoder–decoder) |
| Encoder hidden size | $d_{\mathrm{model}} = 1024$ |
| LoRA rank | $r = 16$ |
| Latent dimension | $d_z = 64$ (per module) |
| Encoder MLP hidden size | 128 |
| Decoder type | Module-aware MoE, 3 heads (enc, dec-self, cross) |
| Experts per head | $E = 4$ |
| Active experts | top-$K = 1$ per location |
| Expert pools | Separate pool per module type (no sharing) |
| Expert MLP input | $2d_z = 128$ (latent + structural embedding) |
| Expert MLP hidden size | 256 (SiLU activation) |
| Expert MLP output size | $r \cdot d_{\mathrm{model}} = 16{,}384$ |
| Gating | Softmax, temperature $\tau = 1.5$ |
| MoE aux loss | Load-balancing, weight $10^{-5}$ |
| Encoder params | $\approx 2.1$M |
| Decoder (MoE) params | $\approx 50.9$M |
| Total generator params | $\approx 53$M ($\approx 7\%$ of backbone) |

## A.5 COMPUTATIONAL COST AND RESOURCE USAGE

We will quantify the practical computational cost of LoRAGen and compare it with the baseline T2L. All experiments for `FLAN-T5-Large` nad `Gemma-2-2B-Instruct` are run on a single NVIDIA A40 GPU (40GB).

**Training time.**  For the in-distribution experiments in Tables 1 and 2, Stage-1 training for T2L and LoRAGen takes about 2 and 2.5 hours, respectively. For the zero-shot experiments in Table 3, the total training time of both methods is less than one day on the same A40 GPU.

**Generator parameter counts.**  For `FLAN-T5-Large`, T2L and LoRAGen (Stage 1) have comparable generator sizes (50M vs. 53M), while LoRAGen already achieves better zero-shot performance. For `Gemma-2-2B-Instruct`, we intentionally use a larger generator for LoRAGen (69M) than for T2L (30M) so that the MoE decoder has enough capacity to generate all four types of LoRA weights, namely qa, qb, va and vb. Notably, the activated parameters of LoRAGen are 15M and 22M per forward pass, respectively.

**Memory usage.**  During LoRA weight autoencoder (LAE) training, enabling the module-aware MoE decoder introduces about 1000 MiB of additional GPU memory, and the adapter-level losses introduce about 300 MiB. In practice, the complete LoRAGen training requires roughly 2500 MiB of extra memory for these components.

## A.6 IMPLEMENTATION DETAILS

In this section we summarize the main architectural and training configurations used in our experiments shown in Tables 7, 8, 9 and 10.

Table 8: Stage 1: LoRA autoencoder configuration for Gemma-2-2B-Instruct.

| Item | Configuration |
|------|---------------|
| Backbone | Gemma-2-2B-Instruct (decoder-only) |
| Encoder hidden size | $d_{\text{model}} = 2304$ |
| LoRA rank | $r = 8$ |
| Latent dimension | $d_z = 64$ (per module) |
| Encoder MLP hidden size | 128 |
| Decoder type | 4-head MoE for $q_a, q_b, v_a, v_b$ |
| $q_a$ output size | $r \times d_{\text{in}} = 8 \times 2304 = 18{,}432$ |
| $q_b$ output size | $d_{q,\text{out}} \times r = 2048 \times 8 = 16{,}384$ |
| $v_a$ output size | $18{,}432$ |
| $v_b$ output size | $d_{v,\text{out}} \times r = 1024 \times 8 = 8192$ |
| Experts per head | $E = 4$ |
| Active experts | top-$K = 1$ per location |
| Expert pools | Separate pool per head |
| Expert MLP input | $2d_z = 128$ |
| Expert MLP hidden size | 256 (SiLU activation) |
| Expert MLP output | Matching head-specific output sizes above |
| Encoder params | $\approx 5.6\text{M}$ |
| Decoder (MoE) params | $\approx 63.7\text{M}$ |
| Total generator params | $\approx 69.3\text{M}$ ($\approx 3\text{–}4\%$ of backbone) |

Table 9: Stage 1: losses and training schedule for both backbones.

| Item | Configuration |
|------|---------------|
| Adapter-level losses | Direction loss $L_{\text{dir}}$ and spectral loss $L_{\text{spec}}$ on $\Delta W$ |
| Loss weights | 1.0 for $L_{\text{dir}}$, 2.0 for $L_{\text{spec}}$ |
| Optimizer | AdamW |
| Learning rate | $1.115 \times 10^{-3}$ |
| Gradient accumulation | Factor 8 |
| Max epochs | 1000 |
| Complexity details | See Appendix A.4 |

## A.7 END-TO-END INFERENCE LATENCY WITH GENERATED ADAPTERS

Once a LoRA is generated, we simply load the weights into the base model; at inference time the runtime is the same as for any standard LoRA fine-tuned model. On an NVIDIA H800, measuring real evaluation runs, we observe the end-to-end latencies per forward pass shown in Table 11.

The small differences are within standard runtime variability; there is no additional per-token overhead induced by LoRAGen itself. All extra cost is incurred once per task when sampling a LoRA adapter set, as discussed in Appendix A.4.

## A.8 SCALABILITY ANALYSIS

In this section, we evaluate the scalability of LoRAGen with respect to both the number of LoRA locations and the LoRA rank. In the experiments, we define the **total generation latency** as the combined time taken for three parts: (1) **Encoding (Enc.)**: The time taken to encode the input task description and map it to the latent space. (2) **Diffusion (Diff.)**: The time taken to perform diffusion sampling over the latent space. (3) **Decoding (Dec.)**: The time taken to decode the latent vector back into the LoRA parameters.

**Scaling with the number of locations.** We evaluate the performance of LoRAGen as the number of LoRA locations increases. We performed scaling tests on `Gemma-2-2B` (rank $r = 8$, $T = 500$ diffusion steps), varying the number of LoRA locations $L$ across values 24, 48, 72, 96 and measured the total latency.

Table 10: Stage 2: latent diffusion over adapter latents.

| Item | Configuration |
|---|---|
| Latent sequence length (FLAN-T5) | $L = 288$ (one token per LoRA location) |
| Latent sequence length (Gemma-2-2B) | $L = 96$ (one token per LoRA location) |
| Denoiser architecture | 1D Transformer |
| Denoiser layers | 4 |
| Model dimension | 256 |
| Input/output dimension | $d_z = 64$ |
| Conditioning | Task embedding + module/layer embeddings |
| Diffusion steps | $T = 500$ |
| Noise schedule | Linear $\beta$ schedule |
| Objective | $v$-prediction with $\ell_2$ loss |
| Optimizer | AdamW |
| Learning rate | $8 \times 10^{-5}$ |
| Batch size | 32 |
| Training steps | 100k |

Table 11: End-to-end latency per forward pass with oracle vs. LoRAGen-generated LoRA adapters on an NVIDIA H800.

| Base model & adapter | Task | Mean latency (ms) |
|---|---|---|
| FLAN-T5-Large, $r = 16$, oracle LoRA | AP-Rec | 42.10 |
| FLAN-T5-Large, $r = 16$, LoRAGen LoRA | AP-Rec | 40.33 |
| Gemma-2-2B, $r = 8$, oracle LoRA | GSM8K | 206.55 |
| Gemma-2-2B, $r = 8$, LoRAGen LoRA | GSM8K | 130.85 |

From these results presented in Table 12, we observe that the **total latency remains low** even as the number of LoRA locations increases. Even for the configuration with **96 locations**, the total latency stays below **0.6 seconds**, which is negligible compared to training a task-specific LoRA from scratch. The **performance remains stable** with negligible fluctuations in accuracy across different $L$ values, further confirming the efficiency of LoRAGen in scaling with the number of locations.

**Scaling with LoRA ranks.** We evaluate the scaling of LoRAGen with respect to LoRA rank $r$. We keep the latent dimension fixed at $d_z = 64$ and reuse the same decoder/denoiser widths. The scaling experiment was conducted on the `Gemma-2-2B` model with $L = 96$ **locations** and $T = 500$ **diffusion steps** for rank $r \in \{8, 16, 32\}$.

As shown in Table 13, the **total generation latency remains stable** even as the LoRA rank increases. The **MoE decoder structure** remains unchanged, with only the **final linear projections** scaling with $r$. The average accuracy shows **negligible fluctuations**, indicating that increasing the LoRA rank does not introduce significant performance degradation. These results suggest that our design is sufficiently flexible to handle larger ranks without requiring a redesign of the architecture.

In summary, the **total generation latency remains low** even as we increase the number of LoRA locations, and **LoRA rank scaling** does not necessitate any architectural changes. The small changes in latency are well within acceptable limits and do not hinder the performance of the model. These findings demonstrate the **scalability** and **efficiency** of LoRAGen in real-world applications.

A.9    TRAINING AND EVALUATION DATASETS FOR ZERO-SHOT GENERATION

As shown in Figure 9 and Figure 10, we conduct zero-shot generation experiment on 136 training tasks and 7 evaluating tasks from FLAN subset. These evaluated tasks are seperate from the training datasets.

Table 12: Latency and accuracy scaling with LoRA locations. **Avg. Acc. (%)** reports the average accuracy on benchmark tasks as in Table 2.

| LoRA locations $L$ | Enc. / Diff. / Dec. latency (ms) | Total gen. latency (ms) | Avg. Acc. (%) |
|---|---|---|---|
| 24 | 0.25 / 563.6 / 5.2 | 569.0 | 72.5 |
| 48 | 0.25 / 566.9 / 5.5 | 572.6 | 72.4 |
| 72 | 0.40 / 562.7 / 5.6 | 568.7 | 72.6 |
| 96 | 0.40 / 569.6 / 5.7 | 575.7 | 72.7 |

Table 13: Latency and accuracy scaling with LoRA rank. **Avg. Acc. (%)** reports the average accuracy on benchmark tasks as in Table 2.

| Rank $r$ | Enc. / Diff. / Dec. (ms) | Total gen. latency (ms) | Avg. Acc. (%) |
|---|---|---|---|
| 8 | 0.40 / 573.5 / 5.7 | 579.6 | 72.7 |
| 16 | 0.49 / 567.5 / 6.2 | 574.2 | 72.8 |
| 32 | 0.64 / 575.6 / 6.5 | 582.7 | 72.7 |

## A.10 TASK DESCRIPTIONS GENERATED BY A LARGE LANGUAGE MODEL

We automate task description generation for each task by leveraging Deepseek[*]. We query its model with carefully constructed prompts that incentivize diversity to facilitate downstream generalization as shown in Figure 11. In particular, we generate 20 descriptions per task. Figure 12 presents representative examples of task descriptions employed in our experiments.

## A.11 DETAILS OF EXPERIMENT SETUP

More details about the diffusion architecture, baseline settings hyperparameter settings, training details and implementation of weight space loss and module-aware MoE decoder can be found in the anonymous repository `https://github.com/tsinghua-fib-lab/LoRAGen`.

## A.12 LLM USAGE

We utilized ChatGPT-4o [†] to refine the content based on our original writing. All revised text was subsequently reviewed and verified by us. The natural language task descriptions we used are generated by DeepSeek [‡]. All code has undergone comprehensive testing to ensure its reliability.

---

[*]https://www.deepseek.com
[†]https://chatgpt.com
[‡]https://www.deepseek.com

## Training Tasks

"lorahub_flan_t5_large-adversarial_qa_dbidaf_based_on",
"lorahub_flan_t5_large-adversarial_qa_dbert_question_context_answer",
"lorahub_flan_t5_large-adversarial_qa_droberta_answer_the_following_q",
"lorahub_flan_t5_large-adversarial_qa_droberta_based_on",
"lorahub_flan_t5_large-adversarial_qa_dbidaf_question_context_answer",
"lorahub_flan_t5_large-adversarial_qa_droberta_tell_what_it_is",
"lorahub_flan_t5_large-adversarial_qa_dbert_tell_what_it_is",
"lorahub_flan_t5_large-adversarial_qa_dbidaf_answer_the_following_q",
"lorahub_flan_t5_large-adversarial_qa_dbert_answer_the_following_q",
"lorahub_flan_t5_large-adversarial_qa_dbert_based_on",
"lorahub_flan_t5_large-adversarial_qa_droberta_question_context_answer",
"lorahub_flan_t5_large-amazon_polarity_user_satisfied",
"lorahub_flan_t5_large-app_reviews_categorize_rating_using_review",
"lorahub_flan_t5_large-app_reviews_convert_to_rating",
"lorahub_flan_t5_large-amazon_polarity_negative_or_positive_tone",
"lorahub_flan_t5_large-amazon_polarity_convey_negative_or_positive_sentiment",
"lorahub_flan_t5_large-amazon_polarity_would_you_buy",
"lorahub_flan_t5_large-amazon_polarity_Is_this_review",
"lorahub_flan_t5_large-yelp_polarity_reviews",
"lorahub_flan_t5_large-glue_wnli",
"lorahub_flan_t5_large-glue_stsb",
"lorahub_flan_t5_large-glue_cola",
"lorahub_flan_t5_large-anli_r1",
"lorahub_flan_t5_large-glue_mrpc",
"lorahub_flan_t5_large-glue_sst2",
"lorahub_flan_t5_large-super_glue_wic",
"lorahub_flan_t5_large-glue_qnli",
"lorahub_flan_t5_large-qasc_qa_with_combined_facts_1",
"lorahub_flan_t5_large-dbpedia_14_pick_one_category_for_the_following_text",
"lorahub_flan_t5_large-dbpedia_14_given_a_choice_of_categories_",
"lorahub_flan_t5_large-dbpedia_14_given_list_what_category_does_the_paragraph_belong_to",
"lorahub_flan_t5_large-wiki_bio_who",
"lorahub_flan_t5_large-qasc_qa_with_separated_facts_4",
"lorahub_flan_t5_large-dbpedia_14_given_a_list_of_category_what_does_the_title_belong_to",
"lorahub_flan_t5_large-qasc_qa_with_separated_facts_5",
"lorahub_flan_t5_large-qasc_qa_with_separated_facts_1",
"lorahub_flan_t5_large-qasc_qa_with_separated_facts_3",
"lorahub_flan_t5_large-race_high_Select_the_best_answer_no_instructions_",
"lorahub_flan_t5_large-quartz_given_the_fact_answer_the_q",
"lorahub_flan_t5_large-web_questions_whats_the_answer",
"lorahub_flan_t5_large-quac",
"lorahub_flan_t5_large-race_middle_Taking_a_test",
"lorahub_flan_t5_large-dream_baseline",
"lorahub_flan_t5_large-sciq_Multiple_Choice_Closed_Book_",
"lorahub_flan_t5_large-quoref_Guess_Answer",
"lorahub_flan_t5_large-ropes_plain_no_background",
"lorahub_flan_t5_large-wiki_qa_found_on_google",
"lorahub_flan_t5_large-quail_context_question_description_answer_id",
"lorahub_flan_t5_large-wiki_hop_original_generate_subject_and_object",
"lorahub_flan_t5_large-duorc_ParaphraseRC_answer_question",
"lorahub_flan_t5_large-quail_context_description_question_answer_text",
"lorahub_flan_t5_large-race_middle_Select_the_best_answer_generate_span_",
"lorahub_flan_t5_large-social_i_qa_Show_choices_and_generate_answer",
"lorahub_flan_t5_large-quoref_Context_Contains_Answer",
"lorahub_flan_t5_large-quartz_read_passage_below_choose",
"lorahub_flan_t5_large-ropes_background_situation_middle",
"lorahub_flan_t5_large-wiqa_effect_with_string_answer",
"lorahub_flan_t5_large-duorc_ParaphraseRC_decide_worth_it",
"lorahub_flan_t5_large-wiki_qa_Topic_Prediction_Question_and_Answer_Pair",
"lorahub_flan_t5_large-wiki_qa_Topic_Prediction_Question_Only",
"lorahub_flan_t5_large-duorc_SelfRC_title_generation",
"lorahub_flan_t5_large-quail_context_question_description_answer_text",
"lorahub_flan_t5_large-race_high_Write_a_multi_choice_question_options_given_",
"lorahub_flan_t5_large-quoref_Given_Context_Answer_Question",
"lorahub_flan_t5_large-quail_no_prompt_text",
"lorahub_flan_t5_large-wiki_qa_Jeopardy_style",
"lorahub_flan_t5_large-wiqa_does_the_supposed_perturbation_have_an_effect",
"lorahub_flan_t5_large-ropes_plain_background_situation",

"lorahub_flan_t5_large-wiki_hop_original_generate_object",
"lorahub_flan_t5_large-kilt_tasks_hotpotqa_complex_question",
"lorahub_flan_t5_large-kilt_tasks_hotpotqa_final_exam",
"lorahub_flan_t5_large-wiqa_what_is_the_final_step_of_the_following_process",
"lorahub_flan_t5_large-ropes_prompt_bottom_hint_beginning",
"lorahub_flan_t5_large-race_middle_Select_the_best_answer",
"lorahub_flan_t5_large-quail_description_context_question_answer_text",
"lorahub_flan_t5_large-social_i_qa_Check_if_a_random_answer_is_valid_or_not",
"lorahub_flan_t5_large-ropes_prompt_mix",
"lorahub_flan_t5_large-ropes_given_background_situation",
"lorahub_flan_t5_large-sciq_Multiple_Choice_Question_First",
"lorahub_flan_t5_large-wiki_hop_original_choose_best_object_affirmative_1",
"lorahub_flan_t5_large-race_middle_Select_the_best_answer_no_instructions_",
"lorahub_flan_t5_large-quoref_Answer_Friend_Question",
"lorahub_flan_t5_large-wiki_hop_original_explain_relation",
"lorahub_flan_t5_large-duorc_SelfRC_question_answering",
"lorahub_flan_t5_large-ropes_prompt_beginning",
"lorahub_flan_t5_large-sciq_Direct_Question_Closed_Book_",
"lorahub_flan_t5_large-race_high_Taking_a_test",
"lorahub_flan_t5_large-quoref_Find_Answer",
"lorahub_flan_t5_large-duorc_ParaphraseRC_extract_answer",
"lorahub_flan_t5_large-wiki_qa_Decide_good_answer",
"lorahub_flan_t5_large-duorc_ParaphraseRC_title_generation",
"lorahub_flan_t5_large-quoref_Found_Context_Online",
"lorahub_flan_t5_large-sciq_Direct_Question",
"lorahub_flan_t5_large-wiki_hop_original_choose_best_object_interrogative_2",
"lorahub_flan_t5_large-wiki_qa_exercise",
"lorahub_flan_t5_large-ropes_background_new_situation_answer",
"lorahub_flan_t5_large-wiki_hop_original_choose_best_object_affirmative_3",
"lorahub_flan_t5_large-quartz_having_read_above_passage",
"lorahub_flan_t5_large-ropes_prompt_bottom_no_hint",
"lorahub_flan_t5_large-quail_context_question_answer_description_text",
"lorahub_flan_t5_large-social_i_qa_Generate_the_question_from_the_answer",
"lorahub_flan_t5_large-duorc_SelfRC_decide_worth_it",
"lorahub_flan_t5_large-quartz_use_info_from_paragraph_question",
"lorahub_flan_t5_large-quail_no_prompt_id",
"lorahub_flan_t5_large-quoref_Read_And_Extract_",
"lorahub_flan_t5_large-race_high_Is_this_the_right_answer",
"lorahub_flan_t5_large-quail_context_question_answer_description_id",
"lorahub_flan_t5_large-wiqa_effect_with_label_answer",
"lorahub_flan_t5_large-web_questions_potential_correct_answer",
"lorahub_flan_t5_large-race_middle_Write_a_multi_choice_question_options_given_",
"lorahub_flan_t5_large-wiqa_which_of_the_following_is_the_supposed_perturbation",
"lorahub_flan_t5_large-duorc_SelfRC_movie_director",
"lorahub_flan_t5_large-wiki_qa_Topic_Prediction_Answer_Only",
"lorahub_flan_t5_large-wiqa_what_might_be_the_last_step_of_the_process",
"lorahub_flan_t5_large-wiqa_what_is_the_missing_first_step",
"lorahub_flan_t5_large-wiki_hop_original_choose_best_object_affirmative_2",
"lorahub_flan_t5_large-duorc_ParaphraseRC_movie_director",
"lorahub_flan_t5_large-duorc_SelfRC_answer_question",
"lorahub_flan_t5_large-wiki_hop_original_generate_subject",
"lorahub_flan_t5_large-quail_description_context_question_answer_id",
"lorahub_flan_t5_large-sciq_Multiple_Choice",
"lorahub_flan_t5_large-quoref_What_Is_The_Answer",
"lorahub_flan_t5_large-social_i_qa_Show_choices_and_generate_index",
"lorahub_flan_t5_large-quoref_Answer_Question_Given_Context",
"lorahub_flan_t5_large-ropes_read_background_situation",
"lorahub_flan_t5_large-duorc_SelfRC_extract_answer",
"lorahub_flan_t5_large-race_high_Select_the_best_answer",
"lorahub_flan_t5_large-para_crawl_enes",
"lorahub_flan_t5_large-newsroom",
"lorahub_flan_t5_large-gem_web_nlg_en",
"lorahub_flan_t5_large-paws_wiki",
"lorahub_flan_t5_large-quarel_testing_students",
"lorahub_flan_t5_large-quarel_logic_test",
"lorahub_flan_t5_large-quarel_choose_between",
"lorahub_flan_t5_large-quarel_do_not_use",
"lorahub_flan_t5_large-quarel_heres_a_story"

Figure 9: Training tasks from FLAN dataset used for training the LoRAGen model

## Validation Tasks

"lorahub_flan_t5_large-amazon_polarity_User_recommend_this_product",
"lorahub_flan_t5_large-amazon_polarity_Is_this_product_review_positive",
"lorahub_flan_t5_large-qasc_is_correct_1",
"lorahub_flan_t5_large-qasc_is_correct_2",
"lorahub_flan_t5_large-wiki_qa_Is_This_True_",
"lorahub_flan_t5_large-wiki_qa_automatic_system"

Figure 10: Validation tasks used during the training the LoRAGen model

# Prompt

## Objective
For every LoRA adapter directory, construct a clean list of task descriptions and turn them into a single sentence-level embedding.
The generator reads curated descriptions from YAML, applies light normalization, matches them by name, and averages T5 sentence embeddings.

## Inputs
- **--yaml_root**: directory with one subfolder per task; each contains `metadata.yaml` with a descriptions list.
- **--logs_root**: Stage-1 checkpoint tree (used to load the VAE encoder for latents; text generation is file-driven).
## Options
- **--strip_generic**: remove leading boilerplate like "The task is / involves / requires …".
-    **--text_pooling {mean, first}**: sentence pooling for T5 (mean is mask-aware and default).

## Procedure
1. **Scan YAML repository**
   For each subfolder under `yaml_root`, load `metadata.yaml` and read descriptions. Skip if missing.
2. **Optional light normalization**
   If `--strip_generic`, strip only the leading boilerplate using a regex and keep the substantive remainder.
3. **Build multi-key alias map**
   Register the cleaned descriptions under:
   - `<entry>`
   - `lorahub_flan_t5_large-<entry>`
   - `lorahub_flan_t5_large_<entry>`
4. **Iterate LoRA adapters**
   For each `…/flan_t5_large_lora/<task_key>/adapter_model.bin`, resolve descriptions by probing name variants (prefix removal and `-`/`_` swaps).
   - **Fallback**: if no hit, use `[task_key]` as the only description and log the miss.
5. **Encode with T5**
   - Tokenize each description, run `T5EncoderModel`.
   - Pool to a sentence vector via:
     - mean pooling (default), or
     - first token pooling.
   - Average across all descriptions → one 1024-d text embedding per task.
6. **Normalize save key**
   Drop `lorahub_flan_t5_large-/_` prefix to form the canonical task name and save:
   ```json
   {
     "task_name": "<canonical_name>",
     "text_embedding": "<1024-d tensor>",
     "latent": "<288 x latent_dim tensor>"
   }
   ```
## Output
One PyTorch file per experiment, e.g.:
   e_000996_with_task_name_vae_task_172_latent_288_64_embed_1024.pt
Mapping canonical task names to averaged text embeddings (plus latents produced in the same pass).

Figure 11: The prompt template used by our pipeline for task descriptions.

**Task descriptions**

**adversarial_qa_dbert_answer_the_followi**
- The task involves reading a passage and answering a question based on the information provided in the text.
- The task requires identifying specific details from a given passage to answer a question.
- The task is about locating exact information within a text to respond to a direct question.

**adversarial_qa_dbidaf_answer_the_following_q**
- The task is to locate the answer to a question within a provided passage.
- The task requires finding the exact words or phrases in a passage that answer a question.
- The task is about answering questions by referring to specific details in a text.

**adversarial_qa_droberta_question_context_answer**
- The task requires matching questions with relevant facts from the given context.
- The task is about locating key details in a text to answer a direct question.
- The task involves reasoning about a passage to derive the correct answer.

**adversarial_qa_dbert_based_on**
- The task involves finding specific information in a given text to answer a direct question.
- The task requires identifying key details from a passage that directly respond to a question.
- The task is about locating exact answers within a provided context based on a question.

Figure 12: Examples of task descriptions generated by our pipeline.

