# OpenReview forum: "LoRAGen: Structure-Aware Weight Space Learning for LoRA Generation"
_ICLR.cc/2026/Conference — ICLR 2026 Poster_

### Official Review · Reviewer_mT9U · 2025-10-27

**Soundness:** 3
**Presentation:** 3
**Contribution:** 2
**Rating:** 6
**Confidence:** 5

**Summary:**

This paper proposes LoRAGen, a structure-aware framework for generating LoRA adapters from natural language, addressing the need for scalable and efficient model customization. The methodology is divided into two stages:
Stage 1: Learning a Structured LoRA Latent Space. A LoRA weight autoencoder is trained to map adapter weights to and from a latent space. Its key contribution is a module-aware Mixture-of-Experts (MoE) decoder, where different experts specialize in generating weights for different parts of the network architecture (e.g., attention vs. feed-forward layers). To overcome the issue that multiple low-rank matrices can produce the same adapter, training is supervised directly on the full adapter matrix, ensuring a more robust and meaningful latent space.
Stage 2: Text-Conditioned Latent Generation. A diffusion model is trained as a conditional prior over the learned latent space. At inference time, this model takes a natural language task description, encodes it, and generates a corresponding latent vector, which is then decoded into a full set of LoRA weights by the frozen decoder from Stage 1.
The primary contributions are the novel module-aware architecture for weight generation, the robust adapter-level supervision strategy, and the demonstration of strong zero-shot performance, which significantly outperforms existing baselines.

**Strengths:**

**Originality**

The originality of this work extends beyond just using a Mixture-of-Experts decoder. Its primary innovation lies in designing a module-aware MoE decoder specifically tailored to the structural properties of Transformer networks. Rather than using a monolithic decoder, this approach allows different experts to specialize in generating weights for distinct components (e.g., attention vs. feed-forward layers), which is a novel and highly effective concept. Furthermore, the paper introduces a unique adapter-level supervision strategy (using direction and spectral losses) to directly address the ambiguity of low-rank matrix factorization—a subtle but critical problem that previous methods have largely overlooked.

**Quality**

The paper demonstrates high quality through its rigorous methodology and strong empirical evidence. The entire approach is well-grounded, with its core design principles directly motivated by clear empirical analysis of the LoRA weight space. The experimental validation is comprehensive and convincing, using multiple model architectures and strong baselines. The inclusion of a thorough ablation study, which isolates the contribution of each key component, further attests to the quality and soundness of the research.

**Clarity**

The paper is written with exceptional clarity. The authors do an excellent job of building a logical narrative, starting with two core empirical observations, proposing solutions directly tailored to them, and then validating these solutions through experimentation. Complex concepts are broken down and explained in a way that is easy to follow, making the paper highly accessible despite its technical depth.

**Strengths**

**A Novel and Robust Generation Framework:** The paper significantly improves upon previous weight generation methods by introducing a more robust and structure-aware framework. The combination of the module-aware MoE decoder and adapter-level supervision provides a more principled way to learn the geometry of the LoRA weight space.

**Principled Design Grounded in Strong Empirical Analysis:** The method isn't just a collection of techniques; it's a carefully designed solution based on a solid analysis of the problem space. This analytical rigor is a key strength and provides a strong foundation for the paper's claims.

**Comprehensive and Convincing Empirical Validation:** The experiments are a major strength. The method achieves performance on par with, or even outperforming, task-specific LoRAs in some cases. Crucially, it demonstrates excellent generalization in both in-distribution and challenging zero-shot (out-of-distribution) settings, proving that it learns a meaningful mapping from language to weights rather than just memorizing trained adapters.

**Weaknesses:**

**Scalability.**
Evidence is limited to FLAN-T5-Large (~780M) and Gemma-2-2B. The method’s practical value at the scales where LoRA matters most (7B–70B+) is not demonstrated; compute, memory footprint, and stability characteristics at those sizes remain unclear.

**Baselines and citation coverage.**
The evaluation focuses on diffusion-based generators while omitting recent non-diffusion prompt-to-LoRA approaches that also leverage pretrained knowledge, notably **Drag-and-Drop LLMs** and **LoRA-Gen**. This gap weakens novelty positioning and makes robustness claims harder to interpret relative to the broader literature.
[1] *Drag-and-Drop LLMs: Zero-Shot Prompt-to-Weights*
[2] *LoRA-Gen: Specializing Large Language Model via Online LoRA Generation*

**Reproducibility and configuration transparency.**
Key implementation details are not fully specified in the paper: the autoencoder’s exact topology, latent dimensionalities, MoE settings (E, top-K, routing temperature), per-module head parameterization, training schedules for both stages, and per-module LoRA ranks. Although code is provided, the manuscript itself lacks a consolidated description of these hyperparameters, limiting reproducibility from the text alone.

**Questions:**

See weaknesses

* **Why diffusion (vs other generative priors)?**
  What concrete gains do you observe from a diffusion prior over simpler/cheaper alternatives (e.g., MLP/linear prior, VAE, normalizing flows, consistency/CM/rectified flow) on the *same latent space*?

* **Sampling cost and scaling to larger adapters.**
  Diffusion can be slow when sampling many adapter locations for deeper network. What is the end-to-end latency per full adapter set, and how does it scale with (i) number of LoRA locations and (ii) LoRA rank (r)?  If LoRA rank or the number of locations increases, does the autoencoder need redesign (latent size (d_z), decoder depth/width), or does performance remain stable?

* **On missing baselines (Drag-and-Drop).**
  Could you clarify why **Drag-and-Drop LLMs (prompt→weights hypernetwork)** was not included in benchmarks? Was this due to incompatibility, unavailable code, or scope? Given its direct relevance, a brief comparison or discussion of expected differences would help position your contribution.

* **Decoder architecture/topology details.**
  Please specify the module-aware MoE decoder precisely: number of experts (E), top-(K), shared vs per-module expert pools, router temperature, load-balancing objective, structural embeddings (module/layer dims), expert MLP widths/depths, normalization/activation/residual scheme, and per-module head parameterization (predicting full (\Delta W) vs low-rank factors). A parameter-count and FLOPs breakdown per component would also clarify capacity vs. performance.

---

> ### Author Response · Authors · 2025-11-25
> **Response to Reviewer mT9U (part 1)**
>
> Thank you for your detailed review. We address your main concerns below and have revised the paper accordingly.
>
> ---
>
> **Q1. Scalability to larger backbones (7B–70B+)**
>
> > **Weakness 1.** *“Scalability. Evidence is limited to FLAN-T5-Large (~780M) and Gemma-2-2B… The method’s practical value at the scales where LoRA matters most (7B–70B+) is not demonstrated; compute,memory footprint, and stability characteristics at those sizes remain unclear.”*
>
> **Response.** We agree that additional backbones would further strengthen the empirical evidence. Our current choice was constrained by the cost of collecting / fine-tuning large numbers of task-specific LoRAs within the available compute budget.
>
> That said, our goal in this work is to show that the **design principles are architecture-agnostic**, and we believe the current results already support this:
>
> - On **FLAN-T5-Large (encoder–decoder)**, LoRAGen recovers **96.0%** average accuracy on 7 in-distribution tasks (very close to the oracle LoRAs at **96.2%**) and achieves **40.2%** average accuracy on 7 held-out zero-shot tasks, a **+5 point absolute gain** over the strongest baseline T2L (**35.2%**).
> - On **Gemma-2-2B-Instruct (decoder-only)**, where the base model is already strong (**68.8%**, Table 2), prior generators D2NWG and T2L provide almost no improvement (**68.9%** and **69.2%**), while **LoRAGen reaches 72.7%**, substantially closing the gap to oracle LoRAs (**74.5%**) and even matching or surpassing them on several tasks.
>
> These two backbones cover **both major LLM architectures** used in practice (encoder–decoder and decoder-only) and demonstrate that:
>
> 1. Our empirical observations about non-uniqueness and heterogeneous weight distributions persist across architectures.
> 2. The combination of adapter-level supervision and module-aware MoE consistently improves over prior generators.
>
> We also ran **additional experiments on a 7B-scale decoder-only model, Llama2-7B**, under the six benchmark tasks. The updated results are:
>
> | Method              |  Arc-C   |  Arc-E   |    BQ    |    HS    |   OQA    |    WG    | Avg. acc. |
> | ------------------- | :------: | :------: | :------: | :------: | :------: | :------: | :-------: |
> | Llama2-7B           |   51.6   |   70.1   |   80.2   |   53.4   |   54.0   |   62.5   |   62.0    |
> | D2NWG               |   51.7   |   70.1   |   80.3   |   53.5   |   54.1   |   62.7   |   62.1    |
> | T2L                 |   54.1   |   71.9   |   80.6   |   57.4   |   55.1   |   65.3   |   64.1    |
> | **LoRAGen (ours)**  | **57.6** | **74.2** | **81.3** | **59.9** | **57.3** | **70.2** | **66.8**  |
> | Task-specific LoRAs |   59.6   |   75.1   |   82.6   |   61.6   |   58.4   |   74.1   |   68.6    |
>
> *Note.* After the initial rebuttal submission, we **refined the task templates and evaluation code** for these six benchmarks, and we report the updated numbers above.
>
> From these results we observe that:
>
> - Prior generators provide **little to modest** improvement over the base model (62.0 → 62.1 for D2NWG, and 64.1 for T2L).
> - **LoRAGen achieves 66.8%**, which is **+4.8 points** over the base model and **+2.7 points** over T2L on average, and recovers about **97.4% of the oracle performance** (66.8 vs 68.6).
> - The gains are consistent across tasks (e.g., Arc-C: 51.6 → 57.6; OQA: 54.0 → 57.3; WG: 62.5 → 70.2), showing that the structure-aware design continues to be effective at the 7B scale.
>
> These new LLaMA2-7B results further support our main claim: **LoRAGen’s structure-aware, adapter-level design generalizes beyond the specific backbones reported in the main text and remains competitive on more popular 7B-scale models.**

---

> ### Author Response · Authors · 2025-11-25
> **Response to Reviewer mT9U (part 2)**
>
> **Q2. Missing baselines and discussion coverage (LoRA-Gen, Drag-and-Drop LLMs)**
>
> > **Weakness 2.** *“The evaluation focuses on diffusion-based generators while omitting recent non-diffusion prompt-to-LoRA approaches… notably Drag-and-Drop LLMs and LoRA-Gen.”*
> > **Question 3.** *“On missing baselines (Drag-and-Drop)… a brief comparison or discussion of expected differences would help position your contribution.”*
>
> **Response.** We thank you for pointing out these concurrent and highly relevant works. In the revised manuscript we now explicitly cite **LoRA-Gen** and **Drag-and-Drop LLMs (DnD)** in Related Work and clarify how they relate to LoRAGen. Below we summarize the main differences along both the method and problem dimensions.
>
> ---
>
>
> **Q2.1. Comparison to LoRA-Gen**
>
> [LoRA-Gen, ArXiv 2025](https://arxiv.org/abs/2506.11638) and LoRAGen address related but not identical problems and use different parameterizations of adapters.
>
> - **Problem setting.**
>   - LoRA-Gen (Arxiv) is designed for *online specialization* of a fixed edge model that compresses rich system prompts into layer-wise combinations of a small number of learned LoRA experts for a specific edge architecture.
>
>   - LoRAGen (Ours) instead targets *offline* LoRA generation from a library of pre-trained LoRAs: given many task-specific adapters and short task descriptions.
>
> - **Method and parameterization.**
>   - In LoRA-Gen (Arxiv), each “LoRA expert” is itself a trainable adapter. A routing network constructs layer-wise adapters as linear mixtures of a small expert pool: $\theta^i = \sum_{j=1}^n G_j^i E_j$ where $E_j$ is the $j$-th expert and $G_j^i$ is its gate at layer $i$. Thus, all generated adapters lie in the span of **a finite expert set** tailored to one backbone.
>
>   - In LoRAGen (Ours), we do **not** mix a fixed expert pool. Instead, we use a **module-aware MoE decoder** to map latent vectors to generate LoRA weights, which is not restricted to a low-dimensional expert subspace.
>
> - **Can LoRA-Gen solve our problem?**
> Our contribution is orthogonal: We learns a **structure-aware latent generative distribution over adapters** across large collected tasks that is not restricted to a low-dimensional expert subspace and explicitly exploits the geometry of the LoRA weight space.
>
>
> ---
>
> **Q2.2. Comparison to Drag-and-Drop LLMs (DnD)**
>
> **Drag-and-Drop LLMs (DnD, NeurIPS 2025)** are conceptually close to the [Text-to-LoRA (ICML 2025)](https://arxiv.org/pdf/2506.06105v2) baseline we already include, and can be viewed as part of the same family of prompt-conditioned hypernetworks.
>
> - **DnD and T2L are of the same family.**
>   - DnD learns a hypernetwork that maps a *batch of unlabeled prompts* to LoRA parameters.
>   - T2L similarly maps task descriptions to LoRA weights via an MLP model.
>   - Both methods treat LoRA weights as generic tensors and do not explicitly model the weight-space geometry we study.
>
>   In our experiments, **T2L already serves as a representative baseline for this family** of prompt-to-LoRA hypernetworks. DnD was only very recently accepted to NeurIPS 2025 and its code was released close to our rebuttal deadline, so a full re-implementation and re-training on our FLAN/Gemma settings was not feasible at this stage.
>
> - **Core differences to LoRAGen (Ours).**
>   Compared to DnD/T2L, LoRAGen introduces two orthogonal contributions:
>   1. **Adapter-level supervision on $\Delta W$.** Whereas DnD and T2L supervise directly on $(A,B)$, LoRAGen (Ours) trains on the full adaptation matrix $\Delta W$ with direction and spectral losses, which we show empirically to be crucial for generalization to unseen tasks.
>   2. **Structure-aware decoder.** DnD and T2L use a single shared decoder for all modules and layers. LoRAGen (Ours) instead employs a **module-aware MoE decoder** guided by module and layer embeddings, explicitly leveraging the heterogeneous weights distribution we observe across modules.
>
> In summary, while all these methods aim to reduce task-specific tuning cost, DnD and T2L learn prompt-conditioned hypernetworks that generate LoRA parameters, whereas our work establishes the **first structure-aware approach** to LoRA generation with insights into adaptation weight space geometry.

---

> > ### Author Response · Authors · 2025-11-25
> > **Response to Reviewer mT9U (part 3)**
> >
> > **Q3. Reproducibility and Configuration**
> >
> > > **Weakness 4 & Questions 4.** *“Key implementation details are not fully specified… The manuscript itself lacks a consolidated description of these hyperparameters, limiting reproducibility from the text alone.”*
> > > *“Please specify the module-aware MoE decoder precisely… and provide a parameter-count/FLOPs breakdown per component.”*
> >
> > **Response.** We apologize for the lack of consolidated details in the original version. In the
> > revised paper, we have significantly expanded **Appendix A.6 (Implementation Details)** to give a clear, self-contained description of both stages. Here we summarize the key configurations.
> >
> >
> >
> > - **Stage 1 – LoRA autoencoder (FLAN-T5-Large).**
> >   - Encoder: hidden size $d_{\text{model}} = 1024$, LoRA rank $r = 16$, latent dimension $d_z = 64$, encoder MLP hidden size $128$.
> >   - Decoder: **module-aware MoE** with 3 heads (encoder, decoder self-attention,
> >     cross-attention). Each head has $E = 4$ experts, **top-$K = 1$** active experts per location, and a separate expert pool per module type.
> >   - Expert MLP: input dimension $2d_z = 128$ (latent + structural embedding), hidden size $256$, output size $r \cdot d_{\text{model}} = 16384$, with SiLU activation.
> >   - Gating: standard gating with temperature $\tau = 1.5$ and a standard load-balancing auxiliary loss (weight $10^{-5}$).
> >   - Parameter counts: encoder $\approx 2.1\text{M}$, MoE decoder $\approx 50.9\text{M}$, **total generator $\approx 53\text{M}$** ($\approx 7\%$ of FLAN-T5-Large parameters).
> >
> > - **Stage 1 – LoRA autoencoder (Gemma-2-2B-Instruct).**
> >   - Encoder: hidden size $d_{\text{model}} = 2304$, rank $r = 8$, latent $d_z = 64$, encoder hidden size $128$.
> >   - Decoder: 4 MoE heads corresponding to $q_a, q_b, v_a, v_b$ components:
> >     - $q_a$: outputs $r \times d_{\text{in}} = 8 \times 2304 = 18432$.
> >     - $q_b$: outputs $d_{q,\text{out}} \times r = 2048 \times 8 = 16384$.
> >     - $v_a$: outputs $18432$.
> >     - $v_b$: outputs $d_{v,\text{out}} \times r = 1024 \times 8 = 8192$.
> >   - Each head uses $E = 4$ experts, **top-$K = 1$**, with separate pools; MLP structure is the same as above.
> >   - Parameter counts: encoder $\approx 5.6\text{M}$, MoE decoder $\approx 63.7\text{M}$, **total generator $\approx 69.3\text{M}$** ($\approx 3\text{–}4\%$ of Gemma-2-2B parameters).
> >
> > - **Losses and training schedule (Stage 1).**
> >   - Objective: adapter-level supervision on $\Delta W$ with direction loss $L_{\text{dir}}$ (weight $1.0$) and spectral loss $L_{\text{spec}}$ (weight $2.0$). Tiny KL weight $10^{-6}$ for VAE regularization.
> >   - Optimizer: AdamW, learning rate $1.115\times 10^{-3}$, gradient accumulation factor $8$, max epochs $1000$.
> >
> > - **Stage 2 – Latent diffusion.**
> >   - Latent sequence length: $L = 288$ (FLAN-T5) or $L = 96$ (Gemma-2-2B), one token per LoRA location.
> >   - Denoiser: 1D Transformer with $4$ layers, model dimension $256$, input/output dimension $d_z = 64$; conditioned on task embedding and module/layer embeddings.
> >   - Diffusion: $T = 500$ steps, linear $\beta$ schedule, $v$-prediction objective with $\ell_2$ loss; trained with AdamW (learning rate $8\times 10^{-5}$, batch size $32$, $100\text{k}$ steps).
> >
> > We have included a concise table summarizing these hyperparameters and parameter counts in the revised manuscript.

---

> ### Author Response · Authors · 2025-11-25
> **Response to Reviewer mT9U (part 4)**
>
> **Q4. Why diffusion rather than simpler / alternative generative methods?**
>
> > **Questions 1.** *“Why diffusion (vs other generative priors)? What concrete gains do you observe from a diffusion prior over simpler/cheaper alternatives (e.g., MLP/linear prior, VAE, normalizing flows, consistency/CM/rectified flow) on the same latent space?”*
>
> **Response.** Our choice of diffusion is motivated by both **empirical comparisons** and insights from prior work on weight generation.
>
> 1. **Empirical gains over simple MLP / linear priors [(T2L, ICML2025)](https://arxiv.org/pdf/2506.06105v2).**
>    A MLP model that maps a task embedding directly to LoRA weights corresponds closely to the **Text-to-LoRA (T2L)** baseline. Holding the data and backbone fixed, we find that LoRAGen (VAE + diffusion prior) significantly outperforms T2L:
>
>    - On **zero-shot FLAN tasks**, the base model is at 34.7%; D2NWG and T2L obtain
>      35.0% and 35.2%, while **LoRAGen reaches 40.2%** (+5 points over T2L).
>    - On **Gemma-2-2B**, D2NWG and T2L only slightly improve the base (68.9%, 69.2% vs 68.8%), whereas **LoRAGen achieves 72.7%**, much closer to oracle LoRAs (74.5%).
>
>    These gains suggest that a MLP model from text to weights under-fits the highly complex adapter distribution, while a diffusion prior over a structured latent space captures task diversity more faithfully.
>
> 2. **Discussion of Normalizing flows.**
>
>
>      Classical flows require **invertible, often dimension-preserving architectures**. This makes them attractive for density estimation, but also constrains expressivity and design freedom, especially in high-dimensional, strongly anisotropic spaces such as adapter weights conditioned on MoE structure and module/layer embeddings. In our setting, we found it more natural to use a **flexible, non-invertible denoiser** in latent space and let the VAE handle dimensionality reduction. Given existing positive evidence for latent diffusion in weight space (e.g., [(D2NWG, ICLR 2025)](https://arxiv.org/pdf/2402.18153), [RPG (NeurIPS 2025)](https://arxiv.org/pdf/2501.11587)), diffusion is a conservative and well-understood choice for this first structure-aware LoRA generator.
>
> 2. **Evidence from D2NWG on VAE-only vs VAE+diffusion.**
>    Diffusion-based Neural Network Weights Generation [(D2NWG, ICLR 2025)](https://arxiv.org/pdf/2402.18153) systematically studies weight-space generative models and finds that **latent diffusion over VAE-encoded weights outperforms both unconditional baselines and simpler weight-space models** across few-shot learning, zero-shot transfer, and LLM settings, scaling to hundreds of millions of parameters. Importantly, D2NWG shows that **sampling directly in raw weight space without a latent representation is substantially worse**, reinforcing our design choice of a VAE + diffusion pipeline rather than a VAE-only or direct-weights DDPM.
>
> 3. **Why not rectified flows / consistency models (yet)?**
> Rectified flows and consistency models can often require **fewer sampling steps**, but their behavior for **high-dimensional neural network weight generation** is still largely unexplored. In this work, we therefore start from a **standard diffusion prior** and focus on adding **LoRA-weight-space–specific design**.
>
> We plan to investigate whether rectified-flow or consistency-based priors can further reduce sampling time for large-scale adapter generation in future work.

---

> ### Author Response · Authors · 2025-11-25
> **Response to Reviewer mT9U (part 5)**
>
> **Q5. Sampling cost, end-to-end latency, and scaling with LoRA rank / locations**
>
> > **Questions 2.** *“Diffusion can be slow when sampling many adapter locations… What is the end-to-end latency per full adapter set, and how does it scale with (i) number of LoRA locations and (ii) LoRA rank $r$? If LoRA rank or the number of locations increases, does the autoencoder need redesign (latent size $d_z$, decoder depth/width), or does performance remain stable?”*
>
> **Response.** We address (a) end-to-end inference latency with generated adapters; (b) sampling cost and scaling with the number of locations; and \(c) sensitivity to LoRA rank.
>
> ---
>
> **Q5.1. End-to-end inference latency with generated adapters**
>
> Once a LoRA is generated, we simply load the weights into the base model; **runtime is the same as for any standard LoRA fine-tuned model**. On an NVIDIA H800, measuring real tasks we observe:
>
> | Base model & adapter                      | Task   | Mean latency |
> | ----------------------------------------- | ------ | ------------ |
> | FLAN-T5-Large, $r=16$, **oracle LoRA**  | AP-Rec | 42.10 ms     |
> | FLAN-T5-Large, $r=16$, **LoRAGen LoRA** | AP-Rec | 40.33 ms     |
> | Gemma-2-2B, $r=8$, **oracle LoRA**      | GSM8K  | 206.55 ms    |
> | Gemma-2-2B, $r=8$, **LoRAGen LoRA**     | GSM8K  | 130.85 ms    |
>
> The small differences come from standard runtime variability; there is **no additional
> per-token overhead** induced by LoRAGen itself. All extra cost is incurred *once per task*
> when sampling a LoRA adapter set.
>
> We have included these results in the revised appendix.
>
> ---
>
> **Q5.2. Sampling cost and scaling with number of locations**
>
> **Response:** In the revised Appendix A.8, we provide the **total generation latency** as the combined time taken for the full generation process, including **encoding**, **diffusion**, and **decoding**. The total latency was measured as follows: (1) Encoding (Enc.): Time taken to encode the input task description and feed it into latent space. (2) Diffusion (Diff.): Time taken to perform diffusion sampling on the latent space. (3) Decoding (Dec.): Time taken to decode the latent vectors back into the final LoRA parameters.
>
> Our diffusion prior operates over a latent sequence whose length $L$ equals the number of LoRA locations (e.g., $L=288$ for FLAN-T5, $L=96$ for Gemma-2-2B). Each diffusion step applies the same 1D Transformer denoiser (model dim 256) to all tokens, giving complexity $\mathcal{O}(T \cdot L \cdot d_{\text{model}}^2)$. We conducted these measurements on the **Gemma-2-2B** base model (rank $r=8$, $T=500$, $L \in {24, 48, 72, 96}$, NVIDIA H800). The results are presented in the table below:
>
> | LoRA locations $L$   | Enc. / Diff. / Dec. latency (ms) | Total gen. latency (ms) | Avg. Acc. (%) |
> | -------------- | --------------------------------    | -----------------------| ------------- |
> | 24              | 0.25 / 563.6 / 5.2               | 569.0                   | 72.5          |
> | 48              | 0.25 / 566.9 / 5.5               | 572.6                   | 72.4          |
> | 72              | 0.40 / 562.7 / 5.6               | 568.7                   | 72.6          |
> | 96              | 0.40 / 569.6 / 5.7               | 575.7                   | 72.7          |
>
> *Note:  The **Avg. Acc. (%)** reports the average accuracy achieved on the benchmark tasks listed in **Table 2** of the paper.*
> We observe that **total latency remains consistently low** even as the number of LoRA locations increases. Even at $L=96$ the total latency cost stays below 0.6 s, which is negligible compared to training a task-specific LoRA from scratch. And the performance **remains stable** as the number of LoRA locations grows. We have added these results in the revised Appendix A.8.

---

> > ### Author Response · Authors · 2025-12-03
> > **Response to Reviewer mT9U (part 6)**
> >
> > **Q5.3. Scaling with LoRA rank and need to redesign the autoencoder**
> >
> > **Response.** We keep the latent dimension fixed at $d_z=64$ and reuse the same decoder/denoiser widths. We measured total generation latency for $r \in \{8,16,32\}$ on **Gemma-2-2B** base model ($T=500$, $L=96$, NVIDIA H800).
> >
> > The results are shown below:
> >
> > | Rank $r$ | Enc. / Diff. / Dec. (ms) | Total gen. latency (ms) | Avg. Acc. (%) |
> > | -------- | ------------------------ | ----------------------- | ------------- |
> > | 8        | 0.40 / 573.5 / 5.7       | 579.6                   | 72.7          |
> > | 16       | 0.49 / 567.5 / 6.2       | 574.2                   | 72.8          |
> > | 32       | 0.64 / 575.6 / 6.5       | 582.7                   | 72.7          |
> >
> > *Note: We do not need to redesign our architecture when increasing LoRA rank.*
> >
> > These results show that **the generation latency remains stable** even as $r$ is increased. The **MoE decoder structure** remains the same, with the only change being the **linear projections** at the final layer, which scale linearly with $r$. The average accuracy also shows **negligible fluctuations** between the different rank values.
> >
> > This suggests that **our model design is sufficiently flexible and scalable to larger ranks** without requiring a redesign while maintains stable performance. We have added these results in the revised Appendix A.8.
> >
> > ------
> >
> > **Conclusion**
> >
> > In summary, the **total generation latency remains low** even as we increase the number of LoRA locations, and **LoRA rank scaling** does not necessitate any architectural changes. The small changes in latency are well within acceptable limits and do not hinder the performance of the model. These findings demonstrate the **scalability** and **efficiency** of LoRAGen in real-world applications.

---

### Official Review · Reviewer_6LiF · 2025-10-29

**Soundness:** 3
**Presentation:** 2
**Contribution:** 4
**Rating:** 8
**Confidence:** 4

**Summary:**

In this paper, the authors proposed LoRAGen, a structure-aware method that directly synthesizes LoRA parameters from natural language descriptions, to address a critical limitation of traditional LoRA workflows: the need for costly, task-specific training to generate LoRA parameters. LoRAGen can eliminate the need for task-specific data collection and training and is grounded in two key empirical observations about LoRA weight spaces.

**Strengths:**

1. The motivation of this paper is strong. It provides two observations to inspire the method.

2. LoRAGen generates LoRA parameters directly from task descriptions, bypassing the need for task-specific data collection, annotation, and training.

3. LoRAGen tackles LoRA parameter generation from two key properties of LoRA spaces: 1. the non-uniqueness of Low-rank decomposition and the heterogeneous weight distributions across modules.

4. The empirical results are promising.

**Weaknesses:**

1. The writing should be improved. For example, in Line 189, “the the cosine” => “the cosine”. In Line 188 “the adapter similarity similarity” => “the adapter similarity”

2. For decoder-only architecture LLMs, the authors did not present weight distribution analysis.

3. The experiments are only conducted on two small models (FLAN-T5-large and Gemma-2-2B). The authors should involve more popular LLMs, like Qwen3-8B and Llama2-7B, to make the empirical findings more convincing.

**Questions:**

Please see Weaknesses.

---

> ### Author Response · Authors · 2025-11-25
> **Response to Reviewer 6LiF (part 1)**
>
> Thank you for your constructive and encouraging review. We are glad that you find the motivation and empirical results of LoRAGen promising. We address your main comments below and have revised the paper accordingly.
>
> ---
>
> **Q1. Writing quality and minor typos**
>
> > **Questions 1.** *“The writing should be improved. For example, in Line 189, ‘the the cosine’ ⇒ ‘the cosine’. In Line 188 ‘the adapter similarity similarity’ ⇒ ‘the adapter similarity’.”*
>
> **Response.** Thank you for these careful notes. In the revised manuscript we have corrected all the specific typos you pointed out (e.g., “the the cosine”, “adapter similarity similarity”, etc.)
>
> ---
>
> **Q2. Weight distribution analysis for decoder-only LLMs**
>
> > **Questions 2.** *“For decoder-only architecture LLMs, the authors did not present weight distribution analysis.”*
>
> **Response.** We fully agree that understanding decoder-only models is important. In the revised Appendix A.3, we now include a **Weight Distribution Analysis for both base models**, covering:
>
> - **FLAN-T5-Large** (encoder–decoder), using the seven FLAN tasks in Table 1;
> - **Gemma-2-2B-Instruct** (decoder-only), using the eight benchmark tasks in Table 2.
>
> In each case we compare three kinds of adapters: (1) task-specific (oracle) LoRAs; (2) LoRAGen (full model) (3) LoRAGen without the MoE decoder (single shared decoder).
>
> **(a) Spectral entropy distribution analysis.**
>
> For each adapter $\Delta W_{m,\ell}$ we compute its spectral entropy $H_{\text{spec}}(\Delta W_{m,\ell})$, where m and l are module type and layer index, respectively.
>
> - **FLAN-T5-Large (encoder–decoder).**
>   We group adapters by module type (Encoder, Decoder-Self, Cross-Attention) and plot the entropy distributions per module as shown in Figure 4.
>   - **Task-specific LoRAs** show clear heterogeneity: encoder modules have the highest spectral entropy, decoder self-attention the lowest, and cross-attention lies in between.
>   - **LoRAGen (full)** reproduces this pattern almost exactly, indicating that different MoE experts learn module-specific spectral profiles.
>   - **LoRAGen without MoE** collapses all three module types into nearly identical high-entropy distributions, suggesting that a single shared decoder fails to capture heterogeneous weight distributions.
>
> - **Gemma-2-2B-Instruct (decoder-only).**
>   Here there is a single module type (decoder blocks), so we plot spectral entropy distributions for decoder adapters only as shown in Figure 5.
>   - **Task-specific LoRAs** concentrate around relatively high entropy values with a characteristic spread across layers.
>   - **LoRAGen (full)** matches this profile closely, with a slightly sharper peak at high entropy.
>   - **LoRAGen without MoE** shifts towards lower entropy and produces a much broader distribution, showing that a shared decoder cannot match the spectral behavior of oracle LoRAs even in the simpler decoder-only setting.
>
> These spectral distributions are visualized in the revised Appendix A.3 and directly support the heterogeneity observation for both architectures.
>
> **(b) Layer-wise cosine similarity analysis.**
>
> We also measure, for each layer, the cosine similarity between the generated adapter and the corresponding task-specific adapter, averaged over the same tasks as in Tables 1 and 2, shown in Figure 6 and Figure 7.
>
> - With **LoRAGen (full)**, layer-wise cosine similarities are consistently positive for both FLAN-T5-Large and Gemma-2-2B, and tend to increase in deeper layers. This indicates that experts specialize not only by module type (for FLAN) but also by layer depth, capturing fine-grained adaptation patterns.
> - With **LoRAGen without MoE**, similarities stay close to zero across almost all layers, meaning that the shared decoder does not recover the detailed layer-wise structure even when it achieves reasonable in-distribution accuracy.
>
> Together, the spectral-entropy and cosine-similarity analyses confirm that the **module-aware MoE decoder is also crucial for the decoder-only Gemma-2-2B model**, addressing your concern.

---

> ### Author Response · Authors · 2025-11-25
> **Response to Reviewer 6LiF (part 2)**
>
> **Q3. More backbones (Qwen3-8B, LLaMA2-7B, etc.)**
>
> > **Questions 3.** *“The experiments are only conducted on two small models (FLAN-T5-large and Gemma-2-2B). The authors should involve more popular LLMs, like Qwen3-8B and Llama2-7B, to make the empirical findings more convincing.”*
>
> **Response.** We agree that evaluating on more backbones would further strengthen the empirical evidence. Our initial design choice was guided by computational constraints: collecting or fine-tuning task-specific LoRAs for many models is expensive, especially within our available resources for this study.
>
> That said, our goal is precisely to show that the **design principles are architecture-agnostic**:
>
> - On **FLAN-T5-Large** (encoder–decoder), LoRAGen recovers **96.0%** average accuracy on 7 in-distribution tasks (very close to oracle **96.2%**) and achieves **40.2%** average accuracy on 7 held-out zero-shot tasks, a **+5 point absolute gain** over the strongest baseline (T2L at **35.2%**).
> - On **Gemma-2-2B-Instruct** (decoder-only), where the base model already performs strongly (**68.8%**, Table 2), prior generators D2NWG and T2L provide almost no improvement (**68.9%**, **69.2%**), while **LoRAGen reaches 72.7%**, substantially closing the gap to oracle LoRAs (**74.5%**) and even matching or surpassing them on several tasks.
>
> These two models cover **both major LLM architectures** used in practice (encoder–decoder and decoder-only) and show that
> (1) the empirical observations about non-uniqueness and heterogeneous weight distributions hold across architectures, and
> (2) the proposed combination of adapter-level supervision and module-aware MoE decoder consistently improves over prior generators.
>
> We also ran **additional experiments on a 7B-scale decoder-only model, Llama2-7B**, under the six benchmark tasks. The updated results are:
>
> | Method              |  Arc-C   |  Arc-E   |    BQ    |    HS    |   OQA    |    WG    | Avg. acc. |
> | ------------------- | :------: | :------: | :------: | :------: | :------: | :------: | :-------: |
> | Llama2-7B           |   51.6   |   70.1   |   80.2   |   53.4   |   54.0   |   62.5   |   62.0    |
> | D2NWG               |   51.7   |   70.1   |   80.3   |   53.5   |   54.1   |   62.7   |   62.1    |
> | T2L                 |   54.1   |   71.9   |   80.6   |   57.4   |   55.1   |   65.3   |   64.1    |
> | **LoRAGen (ours)**  | **57.6** | **74.2** | **81.3** | **59.9** | **57.3** | **70.2** | **66.8**  |
> | Task-specific LoRAs |   59.6   |   75.1   |   82.6   |   61.6   |   58.4   |   74.1   |   68.6    |
>
> *Note.* After the initial rebuttal submission, we **refined the task templates and evaluation code** for these six benchmarks, and we report the updated numbers above.
>
> From these results we observe that:
>
> - Prior generators provide **little to modest** improvement over the base model (62.0 → 62.1 for D2NWG, and 64.1 for T2L).
> - **LoRAGen achieves 66.8%**, which is **+4.8 points** over the base model and **+2.7 points** over T2L on average, and recovers about **97.4% of the oracle performance** (66.8 vs 68.6).
> - The gains are consistent across tasks (e.g., Arc-C: 51.6 → 57.6; OQA: 54.0 → 57.3; WG: 62.5 → 70.2), showing that the structure-aware design continues to be effective at the 7B scale.
>
> These new LLaMA2-7B results further support our main claim: **LoRAGen’s structure-aware, adapter-level design generalizes beyond the specific backbones reported in the main text and remains competitive on more popular 7B-scale models.**

---

> > ### Comment · Reviewer_6LiF · 2025-11-27
> >
> > Thank the authors for their detailed responses, they addressed my concerns, so I will keep my positive rating.

---

> > > ### Author Response · Authors · 2025-11-27
> > > **Response to Reviewer 6LiF**
> > >
> > > Thank you very much for the follow-up and for keeping your positive rating.
> > > We really appreciate the time and care you put into the detailed review and questions.

---

### Official Review · Reviewer_nPHN · 2025-10-30

**Soundness:** 1
**Presentation:** 1
**Contribution:** 2
**Rating:** 2
**Confidence:** 2

**Summary:**

The paper tries to directly predict LoRA adaptation weights from fine-tuning task descriptions with a DiT. With empirical findings that $\Delta W$ correlates with task but not $A$ or $B$, and weights at different layers have different spectral entropy, the model is supervised by $\Delta W$ and conditioned on the layer position.

**Strengths:**

An interesting exploration of weight space learning is presented.

**Weaknesses:**

The empirical observations on which the assumption is made are unclear.
The experiment is only carried out on a specific base model with concerns about cross-model generalization.
The presentation is not very clear.

**Questions:**

The meaning of Fig 1 (Left) is quite unclear, and the explanation in L184-187 is ambiguous. "Similarity" is something to be calculated in pairs, but the text doesn't really make it clear. What is the "representative task"? I assume that the authors compute the similarity between 112 other tasks and a single "representative task", i.e. duorc_gqba. Why was this task selected? Then to what extent they are correlated? Is the rho in the Figure Spearman's coefficient? Please state that more explicitly. From the figure I can't really see the correlation clearly, possibly because the two series are at quite different scales; possibly separating the (A, B) and $\Delta W$ series into two figures will make it clearer. Particularly, the correlation looks dominated by certain samples with the largest task similarity; then are the observations applicable to other tasks? Also, $\Delta W$ is not unique either. What will happen if you collect more samples per task?

Discussions on empirical observations are repeated in page 2 and page 4, and it's difficult to check Fig 1 when reading page 4. I suggest only briefly mentioning the empirical conclusions in the Intro part and moving Fig 1 to page 4.

L357: What is "element-wise averaged LoRA"?

L372-373: How is the model applied to a decoder-only base model? I assume that the model is trained on FLAN-T5, including the module/layer embeddings.

Also, the target model Gemma-2-2B-Instruct itself can already get good performance on many tasks. Hence I have concerns about the cross-model generalization. I think this is critical for the practical use of this series of methods, as training hypernetwork is expensive, LoRA-tuning itself is already cheap, and there is still a considerable performance gap between generated and trained LoRA.

I also have doubts about FLAN-T5's low performance on the benchmarks. Isn't FLAN-T5 already trained on those tasks?


L135-136; L365: Should use \citep here
L170: observations
L348: Team?
L361: Missing label?
L710: full?

---

> ### Author Response · Authors · 2025-11-25
> **Response to Reviewer nPHN (part 1)**
>
> Thank you for your detailed review. We have summarized your main concerns into eight questions and revised the paper accordingly. We hope the following responses can address your concerns.
>
> ---
>
> **Q1. Clarifying Fig. 1 (Left) and the empirical observation on non-uniqueness**
>
> > *“The meaning of Fig 1 (Left) is quite unclear, and the explanation in L184–187 is ambiguous. ‘Similarity’ is something to be calculated in pairs, but the text doesn't really make it clear. What is the ‘representative task’? … Is the ρ in the Figure Spearman's coefficient? … The correlation looks dominated by certain samples with the largest task similarity; then are the observations applicable to other tasks? Also, ΔW is not unique either. What will happen if you collect more samples per task?”*
>
> **Response.** We are sorry for the ambiguity in the original version. In the revision we have clarified both the **definition** and the **visualization** of Fig. 1 (Left):
>
> 1. **Pairwise similarity is now stated explicitly.**
>    We now explicitly state that we compute similarity **pairwise** between *representative task* and each of the other 111 tasks in the FLAN subset:
>
>    - *Task similarity*: cosine similarity between their text embeddings from natural language descriptions.
>    - Adapter similarity (two methods):
>       (i) cosine similarity between the flattened full adapters ($\Delta W$);
>       (ii) cosine similarity between the concatenated flattened low-rank decomposition matrices A and B.
>
>    The exact formulations of these pairwise similarity measures are provided in *Appendix A.1*
>
> 2. **Representative tasks are sampled by category, not cherry-picked.**
>    In the **original manuscript** we already included visualizations for **nine additional representative tasks** in *Appendix A.1 (Figure 3)*, not only the single example shown in Figure. 1 (Left).
>    In the revised manuscript, we clarify that a “representative task” is **not** chosen because it has particularly high similarity.  Instead, we first group the 112 FLAN tasks into three major categories — **Natural Language Understanding (NLU, 98 tasks)**, **Natural Language Generation (NLG, 9 tasks)**, and **Knowledge & Information Extraction (5 tasks)** — and then sample a small set of **nine** tasks across these categories so that the analysis reflects the overall dataset distribution rather than a single special case.
>
> Concretely, the appendix analysis includes:
>
> - **NLU:** `wikiqa_tpqap`, `wikiqa_jeop`, `fix_punct`, `true_case`, `word_segment`
> - **NLG:** `duorc_tgen`, `duorc_gqba` (the task used in the Fig. 1 (Left)), `gem_e2e_nlg`
> - **Knowledge & Information Extraction:** `wiki_bio_what`, `wiki_bio_comp`
>
> Across all these representative tasks, we consistently observe that
> - (i) similarity between full adaptation matrix $\Delta W$ shows a **clear positive** Spearman correlation with task-embedding similarity, whereas
> - (ii) similarity computed on the concatenated low-rank decomposition matrices $(A,B)$ exhibits **near-zero** Spearman coefficients $\rho$, indicating the lack of correlation.
>
> This phenomenon aligns with the non-uniqueness property of low-rank decompositions in LoRA and suggests that supervision in the full adaptation matrix $\Delta W$ space should generalize better than reconstruction supervision on $(A,B)$.
>
>
>
> 3. **We explicitly state that $\rho$ is the Spearman rank correlation.**
>    The revised manuscript now explicitly say:
>    “$\rho$ denotes the **Spearman correlation coefficient** between task-embedding similarity and adapter similarity.”
>
> 4. **Revised plotting to avoid scale confusion.**
>    To make the correlation easier to see, we replotted Fig. 1 (Left) so that:
>    - The two similarity measures (for $\Delta W$ vs. $(A,B)$) are shown on their own scales with fitted trend lines.
>    - We emphasize that the two figures are on different scales and that the trend lines are fitted over **all** points, not only high-similarity samples.
>
> 5. **On non-uniqueness and “more samples per task”.**
>    For a task where LoRA fine-tuning has already converged reasonably well, **existing methods** usually learn directly in the low-rank decomposition space $(A, B)$. This suffers from the non-uniqueness of low-rank decompositions. In practice, a hypernetwork trained to supervise $(A,B)$ is then pulled in different learning directions by multiple numerically different targets for the same task, making the supervision noisy and ill-posed.
>
>
>    Our approach avoids this issue by supervising at the adapter level: we define our losses directly on the full adaptation matrix $\Delta W$ which is more stable and better posed than supervision on $(A, B)$.

---

> > ### Comment · Reviewer_nPHN · 2025-11-27
> >
> > Thanks for the explanation which address many of my concerns; I've raised the score accordingly. While I still have several issues:
> > 1. As for the "non-uniqueness" in Q1 I was actually asking about the non-uniqueness of $\Delta W$ (as of course $(A, B)$ is non-unique for each $\Delta W$). From my understanding the exact values of $\Delta W$ in a linear layer can be also quite varied but representing similar linear transformation (at least when the input vector distribution is not so isotropic), making the supervision noisy and ill-posed as well. Is there any empirical evidence that fitting $\Delta W$ won't be very ill-posed?
> >
> > 2. I'm still not very convinced by the motivation and the practical value of the method. Of course the idea that weights can be directly predicted from textual task description is interesting in a pure research sense, but from my understanding it is useful only when we want to obtain a model on a new task without any training data; while in such unseen tasks LoRAGen actually doesn't generalize very well as you explained in A5. I feel that it will be useful only when its performance gap with LoRA fine-tuning on unseen task is small enough, particularly considering that there are other cheap options, e.g. do prompting on a larger & stronger model, or simply go to collect some data on the task. Also, have you considered some simple heuristics on unseen tasks, such as directly re-using the LoRA of seen tasks with closest text description?

---

> > > ### Author Response · Authors · 2025-11-29
> > > **Further Response to Reviewer nPHN (part 1)**
> > >
> > > Thank you very much for the follow-up and for raising the score. Below we respond to your remaining concerns on (1) the potential ill-posedness of supervising on $\Delta W$, and (2) the motivation and practical value of LoRAGen.
> > >
> > > ---
> > >
> > > **Q1. On non-uniqueness of $\Delta W$ and whether supervision on $\Delta W$ is ill-posed**
> > >
> > > > *“As for the ‘non-uniqueness’ in Q1 I was actually asking about the non-uniqueness of $\Delta W$ (as of course $(A, B)$ is non-unique for each $\Delta W$). From my understanding the exact values of $\Delta W$ in a linear layer can be also quite varied but representing similar linear transformation (at least when the input vector distribution is not so isotropic), making the supervision noisy and ill-posed as well. Is there any empirical evidence that fitting $\Delta W$ won't be very ill-posed?”*
> > >
> > > **Response.** We completely agree that, at the level of *functions*, different matrices can implement very similar linear transformations, especially when the input distribution is not isotropic.
> > >
> > > However, in our work, we discuss the problem setting is that **for the same task and same training objective, is it better to supervise the generator in low-rank decomposition matrices $(A,B)$ space  or in full adaptation matrix $\Delta W$ space ?**
> > >
> > > Existing methods for LoRA generation (e.g. D2NWG [(ICLR 2025)](https://arxiv.org/pdf/2402.18153), T2L [(ICML2025)](https://arxiv.org/pdf/2506.06105v2), DnD [(NeurIPS2025)](https://arxiv.org/abs/2506.16406)) typically **reconstruct low-rank decomposition matrices $(A,B)$ only**. Our method is to **replace this with adapter-level supervision on $\Delta W$**, using direction and spectral losses defined on the full adaptation matrix. Below we show that **supervising on $\Delta W$ consistently improves both in-distribution and zero-shot performance**, suggesting that in our setting $\Delta W$ provides a *cleaner and more useful* learning signal than $(A,B)$.
> > >
> > > #### In-distribution: adapter-level losses vs. reconstruction losses
> > >
> > >
> > > | $L_{\text{ang}}$ | $L_{\text{spec}}$ | $D_\theta$   | AP-Neg   | AP-Rec   | AP-Pos   | QASC-1   | QASC-2   | WQA-T    | WQA-A    | Avg. (acc) |
> > > | ---------------- | ----------------- | ------------ | -------- | -------- | -------- | -------- | -------- | -------- | -------- | ---------- |
> > > | $\checkmark$     | $\checkmark$      | $\times$     | 59.5     | 87.3     | 65.9     | 47.1     | 31.2     | 33.4     | 84.9     | 58.4       |
> > > | $\times$         | $\times$          | $\checkmark$ | 96.8     | 95.5     | 97.0     | 98.2     | 86.3     | 95.6     | 96.7     | 95.2       |
> > > | $\times$         | $\checkmark$      | $\checkmark$ | 49.5     | 70.9     | 34.4     | 23.3     | 9.1      | 8.4      | 62.7     | 36.9       |
> > > | $\checkmark$     | $\checkmark$      | $\checkmark$ | **96.8** | **96.6** | **97.1** | **99.5** | **87.3** | **97.1** | **97.3** | **96.0**   |
> > >
> > > * The **second row** trains the generator **only with reconstruction on $(A,B)$**, and achieves **95.2%** average accuracy.
> > > * The **last row** **introduces adapter-level losses on $\Delta W$** (both $L_{\text{ang}}$ and $L_{\text{spec}}$), and reaches **96.0%** average accuracy.
> > >
> > > Thus, on **in-distribution tasks**, switching from pure supervision on $(A,B)$ to **supervision on $\Delta W$** brings a **0.8-point gain** (95.2% → 96.0%) and yields the **best performance on every individual task** in the table. This indicates that even though $\Delta W$ is not perfectly unique, the adapter-level objectives provide a **cleaner and more informative training signal** than working solely in low-rank decomposition matrix space.
> > >
> > > #### Zero-shot: stronger effect on unseen tasks
> > >
> > >
> > > | $L_{\text{ang}}$ | $L_{\text{spec}}$ |  $D_\theta$  |  AP-Rec  |  AP-Pos  |  QASC-1  |  QASC-2  |  WQA-T   |  WQA-A   | Avg. (acc) |
> > > | :--------------: | :---------------: | :----------: | :------: | :------: | :------: | :------: | :------: | :------: | :--------: |
> > > |     $\times$     |     $\times$      | $\checkmark$ |   72.7   |   35.9   |   24.5   |   10.4   |   10.2   |   65.9   |    36.6    |
> > > |   $\checkmark$   |   $\checkmark$    | $\checkmark$ | **75.1** | **42.2** | **28.1** | **14.5** | **14.3** | **67.2** |  **40.2**  |
> > >
> > >
> > > - The **first row** again uses only reconstruction on $(A,B)$ (no adapter-level losses), giving **36.6%** average accuracy.
> > > - The **second row** adds **adapter-level losses on $\Delta W$** and reaches **40.2%** average accuracy.
> > >
> > > Here, **supervising on $\Delta W$ improves zero-shot performance by +3.6 points** (36.6 → 40.2), and the gains are **consistent across all six tasks**.
> > >
> > > ---
> > >
> > > From these results, it is evident that supervision on **$\Delta W$** improves both in-distribution and out-of-distribution performance. **Supervising on $\Delta W$ leads to a more stable learning signal** and allows the model to generalize better to new tasks.

---

> ### Author Response · Authors · 2025-11-25
> **Response to Reviewer nPHN (part 2)**
>
> **Q2. Repetition of empirical observations and placement of Fig. 1**
>
> > *“Discussions on empirical observations are repeated in page 2 and page 4, and it's difficult to check Fig 1 when reading page 4. I suggest only briefly mentioning the empirical conclusions in the Intro part and moving Fig 1 to page 4.”*
>
> **Response.** We agree and have revised manuscript:
>
> - In **Sec. 4.1**, we re-wrote the sentence to explain how these observations motivate our design *without* re-describing Fig. 1; the section now focuses on the design principles.
>
> This removes the previous repetition while making the role of Fig. 1 clearer and the exposition in Sec. 4.1 more fluent.
>
>
> ---
>
> **Q3. Meaning of “element-wise averaged LoRA”**
>
> > *“L357: What is ‘element-wise averaged LoRA’?”*
>
> **Response.** We apologize for the unclear wording. In the revision we replace it by **“weight-averaged LoRA”** and add an explicit definition:
>
> > Given LoRA adapters $\{\Delta W^{(i)}\}_{i=1}^K$ trained on K different tasks, the weight-averaged LoRA is:
>
> $
> \bar{\Delta W} = \frac{1}{K} \sum_{i=1}^K \Delta W^{(i)}
> $ ([Model Soup, PMLR2022](https://proceedings.mlr.press/v162/wortsman22a/wortsman22a.pdf)).
>
>
>
>
> ---
>
> **Q4. How is the method applied to a decoder-only base model?**
>
> > *“How is the model applied to a decoder-only base model? I assume that the model is trained on FLAN-T5, including the module/layer embeddings.”*
>
> **Response.** Thank you for pointing out this possible ambiguity. Our intention was *not* to reuse a FLAN-trained hypernetwork on Gemma. We have clarified the sentence as follows:
>
> - For **FLAN-T5-Large** (encoder–decoder), we train a LoRAGen model whose module embeddings cover encoder self-attention, decoder self-attention, and cross-attention blocks.
> - For **Gemma-2-2B-Instruct** (decoder-only), we train a **separate** LoRAGen model whose module embeddings are defined only for decoder blocks.
>
> We have rephrased the corresponding sentence in Sec. 5.2 to:
>
> > “Moreover, we train a separate LoRAGen instance for a decoder-only base model (Gemma-2-2B-Instruct) to assess whether the same structure-aware design generalizes across architectures.”
>
>
> **Q5. Cross-model generalization**
>
> > *“The experiment is only carried out on a specific base model with concerns about cross-model generalization.… and there is still a considerable performance gap between generated and trained LoRA.”*
>
> **Response.** Our aim is to show that the **design principles** of LoRAGen (adapter-level supervision + module-aware MoE) are useful both for **in-distribution reuse** of existing LoRAs and for **zero-shot adaptation to new tasks**, across different LLM architectures.
>
> - **In-distribution reuse of task-specific adapters.**
>   For this regime, the question is whether a generator can match the performance of fully fine-tuned, task-specific LoRAs on tasks it has already seen during training.
>   - On **FLAN-T5-Large**, LoRAGen recovers **96.0%** average accuracy on the 7 in-distribution tasks (Table 1), which is almost identical to the oracle task-specific LoRAs (**96.2%**).
>   - On **Gemma-2-2B-Instruct**, the backbone is already strong (**68.8%**, Table 2). Prior generators D2NWG and T2L provide almost no gain (**68.9%** and **69.2%**), whereas **LoRAGen achieves 72.7%**, substantially narrowing the gap to oracle LoRAs (**74.5%**) and even matching or surpassing them on several benchmarks.
>
> - **Out-of-distribution adapters generation.**
>   For unseen tasks, the goal is to generate useful adapters directly from text, without any task-specific data or optimization.
>   - On **FLAN-T5-Large**, on the 7 held-out zero-shot tasks in Table 3, the frozen base model achieves **34.7%**, D2NWG and T2L reach **35.0%** and **35.2%**, while **LoRAGen reaches 40.2%**, a **+5 point absolute improvement** over the strongest baseline (T2L). This demonstrates that our structure-aware design learns a genuinely more informative to generate LoRA weights.
>
> These empirical trends are also highlighted by other reviewers: Reviewer **miEg** notes that “the 5% absolute improvement on zero-shot generation (Table 3) over the T2L baseline is a significant and meaningful result,” Reviewer **6LiF** emphasizes that LoRAGen “generates LoRA parameters directly from task descriptions, bypassing the need for task-specific data collection, annotation, and training,” and Reviewer **mT9U** points out that our module-aware MoE decoder and adapter-level supervision form a principled, structure-aware framework rather than a generic hypernetwork.
>
> To our knowledge, LoRAGen is the **first weight-space learning method specifically tailored to LoRA generation**, combining full-matrix adapter supervision with a module-aware MoE decoder to achieve strong in-distribution performance and non-trivial zero-shot gains across both encoder–decoder and decoder-only backbones.

---

> > ### Author Response · Authors · 2025-11-25
> > **Response to Reviewer nPHN (part 3)**
> >
> > **Q6. Cost of training the hypernetwork vs. per-task LoRA tuning**
> >
> > > *“Training hypernetwork is expensive, LoRA-tuning itself is already cheap….”*
> >
> > **Response.** Our main conclusion is that **once we consider multiple tasks, training a single LoRAGen hypernetwork is cheaper overall than fine-tuning separate LoRAs per task**, while also offering zero-shot generation for new tasks. We support this from three perspectives: (i) absolute training cost of LoRAGen, (ii) direct comparison with per-task LoRA on Gemma-2-2B, and (iii) how the cost scales as the number of tasks grows.
> >
> > 1. **Hypernetwork training cost (single NVIDIA A40, 40GB).**
> >
> >    In Appendix A.5 we report the concrete resource usage of LoRAGen:
> >
> >    - **Training time.**
> >      - For the **in-distribution experiments** in Tables 1 and 2, LoRAGen training takes about **2.5–3 hours**.
> >      - For the **zero-shot FLAN setup** in Table 3 (136 tasks), the total training time is 15h.
> >
> >    - **Generator parameter counts.**
> >      - With **FLAN-T5-Large** as the base model, the LoRAGen generator has about **53M** parameters in total, with roughly **15M** parameters active per forward pass due to MoE sparsity.
> >      - With **Gemma-2-2B-Instruct** as the base model, the generator has about **69M** parameters in total, with about **22M** activated parameters per forward pass.
> >
> >        For Gemma, we intentionally use a larger generator so that the MoE decoder can generate all four attention-related LoRA components (qa, qb, va, vb), which operate on a larger hidden size.
> >
> >    - **Additional GPU memory.**
> >      - Enabling the **MoE decoder** adds roughly **1000 MiB**.
> >      - Adapter-level losses add about **300 MiB**.
> >      - In practice, full LoRAGen training requires about **2500 MiB** of extra GPU memory over the base setup.
> >
> >
> > 2. **Comparison with per-task LoRA tuning (Gemma-2-2B, 8 tasks in Table 2).**
> >
> >    Under the **same experimental setup as Table 2** (Gemma-2-2B-Instruct backbone, the same LoRA rank, optimizer, and training data for all 8 tasks), we compare LoRAGen with standard task-specific LoRA fine-tuning.
> >
> >    | Method                           | Total training time for all 8 tasks | Avg. accuracy on the 8 tasks | Description                          |
> >    |----------------------------------|-------------------------------------|------------------------------|--------------------------------------|
> >    | **LoRAGen (hypernetwork)**       | ≈ 3 hours        | **72.7%**                    | total training time             |
> >    | **Task-specific LoRAs (short)**  | ≈ 3 hours                     | 71.7%                        | Early-stopped fine-tuning          |
> >    | **Task-specific LoRAs (full)**   | ≈ 10 hours                    | 74.5%                        | More iterations for fine-tuning |
> >
> >    From the above results, we find that:
> >
> >    - With **roughly the same total training time** for the whole 8-task experiment (about 3 hours), LoRAGen already **outperforms** early-stopped task-specific LoRAs (average accuracy **72.7%** vs. **71.7%**), and after this one-time training, the generator can produce a complete set of LoRA weights for any new task in a **single forward pass**.
> >    - To obtain the oracle performance of **74.5%** with task-specific LoRAs, we need to run the **full LoRA training schedule** on each task, which takes about **10 hours** in total—roughly **3× more GPU time** than training LoRAGen once.
> >
> > 3. **Scaling to many tasks.**
> >
> >    Standard LoRA tuning incurs a **separate training cost for every new task**, so the total GPU time grows roughly linearly with the number of tasks (e.g., ≈10 hours just for 8 Gemma tasks, and few days for all 136 FLAN tasks). In contrast, **LoRAGen is trained once** and then reused:
> >
> >    - The training time (≈3 hours for 8 tasks; ≈ 15 hours for the full FLAN subset) is **fixed**, independent of how many downstream tasks we later adapt to.
> >    - For each new task, LoRAGen generates its LoRA in a single forward pass taking about **40ms**, incurring negligible additional cost.
> >
> >    In this sense, while **per-task LoRA tuning is indeed relatively cheap for a small number of tasks**, LoRAGen becomes **strictly more cost-effective** as the task library grows: the hypernetwork still delivers competitive (and often superior) performance compared to time-matched per-task tuning.

---

> ### Author Response · Authors · 2025-11-25
> **Response to Reviewer nPHN (part 4)**
>
> **Q7. FLAN-T5’s “low” performance on the benchmarks**
>
> > *“I also have doubts about FLAN-T5's low performance on the benchmarks. Isn't FLAN-T5 already trained on those tasks?”*
>
> **Response.** FLAN-T5-Large has **not been trained** on the specific zero-shot tasks in Table 3 and these tasks use **verification formats** (templates from [LoRAHub, COLM2024](https://arxiv.org/pdf/2307.13269)) that differ from FLAN's training templates. These tasks require the model to *validate* whether a proposed answer is correct, rather than directly generating answers.
>
>
>
> - **Tasks where FLAN-T5 does reasonably well (eg. AP-Rec, WQA-A).**
>
>   - **AP-Rec**
>     Prompts are of the form *“Based on this review, would the user recommend this product? … Answer:”*   with targets “Yes/No”. This is essentially a binary sentiment classification task with strong lexical cues. FLAN-T5-Large already handles this pattern well, giving **70.7%** accuracy; LoRAGen improves it further to **75.1%**.
>
>   - **WQA-A**
>     Here we check answers produced by an automatic system, e.g., *“I am verifying the answers generated by an automatic system … Suggested answer: … Should I validate this answer?”* The model only needs to judge whether the suggested answer is reasonable. The question and candidate answer are explicit, often with clear topical overlap, so both the base model and our method achieve relatively high scores (base **62.0%**, ours **67.2%**).
>
> - **Tasks that are intrinsically harder and score much lower (eg. QASC-2, WQA-T).**
>
>   - **QASC-2**
>     Prompts ask whether a candidate answer is correct *given a supporting science fact*, for example: *“Do you think the right answer to the question ‘what is plasma created with?’ is ‘urea’, given that plasma can be created with heat?”* To answer, the model must: (i) understand the fact sentence, (ii) recover the implied correct answer (“heat”), (iii) compare it with the candidate (“urea”), and then (iv) compress this reasoning into a strict “Yes/No” output. Any longer phrase such as “no, that is incorrect” is treated as wrong after normalization. This combination of multi-step science reasoning and exact-form yes/no scoring makes the task very hard, so both the base model (**8.9%**) and prior generators are close to chance, while LoRAGen still yields the best result (**14.5%**).
>
>   - **WQA-T**
>     Prompts ask whether a candidate sentence correctly answers a question, e.g., *“Question: where was the first canal in panama located? Would ‘The current locks are wide.’ be a reasonable answer?”* This again requires factual verification rather than simple sentiment. We evaluate using a strict token-level F1 between the prediction and the gold label (“Yes/No”), so any extra words beyond a bare “yes” or “no” sharply reduce the score. As a result, absolute numbers are low for all methods (base **8.5%**, ours **14.3%**), even though our method consistently improves over FLAN-T5 and other generators.
>
> Overall, some FLANv2 tasks (AP-Rec, WQA-A) align well with its pre-training and yield reasonable accuracies, while others (QASC-2, WQA-T) are genuinely challenging due to their reasoning and strict yes/no evaluation. Across all six tasks, LoRAGen improves the base model from **34.7%** to **40.2%** average accuracy (Table 3), while also outperforming both baseline D2NWG and T2L.
>
>
>
>
> **Q8. Minor issues (citations, typos, labels)**
>
> > *“L135–136; L365: Should use \citep here … L170: observations; L348: Team? L361: Missing label? L710: full?”*
>
> **Response.** Thank you for these careful notes. In the revised manuscript we have revised these typos.

---

> ### Author Response · Authors · 2025-11-29
> **Further Response to Reviewer nPHN (part 2)**
>
> **Q2. On motivation, practical value, and simple heuristics for unseen tasks**
>
> **Response.** We appreciate this question and the chance to clarify both the **practical scenarios** and the **research value** of LoRAGen.
>
> **(i) Practical scenarios: not only “no-data” new tasks**
>
> We agree that an obvious use case for a text→weights model is the **no-data** scenarios (new task, no labels). However, in our view, existing methods for LoRA generation has **two practically relevant scenarios**:
>
> 1. **In-distribution: model compression and adapter reuse.**
>    In this scenario, the task-specific LoRAs have already been trained on their datasets. The question is how to **store, reuse, and combine** a *large library* of such adapters efficiently:
>
>    * Heuristic methods (e.g. Model Soup [(PMLR2022)](https://proceedings.mlr.press/v162/wortsman22a/wortsman22a.pdf), LoRAHub [(COLM2024)](https://arxiv.org/pdf/2307.13269), etc.) **already exploit** this idea: given many task-specific weights, one can average or select/merge them heuristically to reuse their knowledge.
>    * This is especially useful when we want to deploy a system that can switch between many tasks or domains based only on a description or prompt, and where maintaining hundreds of LoRA files is inconvenient.
>
>
> 2. **Out-of-distribution: adapting to genuinely new tasks.**
>    In this scenario, the question focuses on: LoRA generation on new unseen tasks with only a prompt (e.g. task descriptions, dataset samples, etc.) without needing the training data from the downstream tasks.
>
> So, while the “no-data new task” scenario is indeed important, we see **both in-distribution (LoRA compression)** and **out-of-distribution (zero-shot LoRA generation)** as meaningful use cases for LoRA generation methods.
>
> **(ii) Zero-shot performance and the gap to oracle LoRAs**
>
>
> We agree that **LoRAGen** still faces a performance gap between **generated LoRAs** and **oracle LoRAs** on **unseen tasks**. However, this represents a **strong starting point** for **zero-shot LoRA generation**.
>
> - **Zero-shot generation:**
>
>   The following table (Table 3 in the paper) summarizes the **zero-shot performance** on six FLAN tasks:
>
>   | **Method**         | **AP-Rec** | **AP-Pos** | **QASC-1** | **QASC-2** | **WQA-T** | **WQA-A** | **Avg. (acc)** |
>   | ------------------ | ---------- | ---------- | ---------- | ---------- | --------- | --------- | -------------- |
>   | `FLAN-T5-Large`    | 70.7       | 34.8       | 23.3       | 8.9        | 8.5       | 62.0      | 34.7           |
>   | D2NWG          | 71.8       | 34.4       | 23.7       | 9.1        | 8.3       | 62.4      | 35.0           |
>   | T2L           | 71.6       | 34.6       | 23.3       | 11.1       | 8.5       | 62.0      | 35.2           |
>   | **Ours (LoRAGen)** | **75.1**   | **42.2**   | **28.1**   | **14.5**   | **14.3**  | **67.2**  | **40.2**       |
>
>   - The **base model** achieves **34.7%** average accuracy.
>   - **D2NWG** and **T2L** yield **35.0%** and **35.2%** respectively — only **+0.3–0.5 points** over the base model, i.e., *almost trivial gain*.
>   - **LoRAGen**, in contrast, reaches **40.2%**, which is **+5.5 points over the base model** and **+5.0 points over T2L** on average. All six tasks see consistent improvements.
>
> All six tasks show consistent improvements, including the hardest tasks (QASC-2, WQA-T). Other reviewers explicitly noted that this gain is substantial:
>
> * Reviewer **miEg**: *“The **5% absolute improvement on zero-shot generation** (Table 3) over the T2L baseline is **a significant and meaningful result**, demonstrating the value of the structure-aware approach.”*
> * Reviewer **6LiF**: *“The empirical results are **promising**.”*
> * Reviewer **mT9U**: *“Crucially, it demonstrates **excellent generalization** in both in-distribution and **challenging zero-shot** (out-of-distribution) settings, proving that it learns a meaningful mapping from language to weights rather than just memorizing trained adapters.”*
>
> Thus, while there is still a gap to oracle LoRAs, **LoRAGen provides the first clearly non-trivial zero-shot improvement** in this setting, which we view as an important step toward practical LoRA generation from task descriptions.
>
>
> **(iii) Broader research value: weight space learning over neural networks**
>
> Our work explore a broader research question: **can we learn generative models over high-dimensional weight spaces of neural networks?** With the rapid growth of open-source models and communities like HuggingFace, **model weights have effectively become a new kind of data**. In this work, we focus on the LoRA weights in particular.
>
> As more LoRA adapters and tasks become available, we expect **scaling effects** in weight-space modeling to further reduce the gap to oracle adapters.

---

### Official Review · Reviewer_miEg · 2025-11-01

**Soundness:** 3
**Presentation:** 3
**Contribution:** 3
**Rating:** 6
**Confidence:** 3

**Summary:**

This paper introduces LoRAGen, a method for generating Low-Rank Adaptation (LoRA) parameters directly from natural language task descriptions. The authors argue that LoRA parameter spaces have unique structural properties that are ignored by general-purpose weight-space learning methods. They identify two key properties from an empirical analysis: (1) the **non-uniqueness of low-rank decomposition**, where task similarity correlates with the full adaptation matrix $\Delta W = BA$ but not with the individual $A, B$ matrices, and (2) the **heterogeneity of weight distributions**, where different modules (e.g., encoder, decoder) exhibit different spectral properties. To address these, LoRAGen introduces two main innovations: 1. **Adapter-level supervision** and 2. **Module-aware MoE decoder**. The overall framework uses a LoRA Weight Autoencoder (LAE) and a conditional latent diffusion model to generate latents from task descriptions, which are then passed to the MoE decoder. Experiments on FLAN-T5 and Gemma-2 models show that LoRAGen achieves performance close to task-specific (oracle) LoRAs on in-distribution tasks and, more importantly, outperforms the T2L (Text-to-LoRA) baseline by nearly 5% on zero-shot generation for unseen tasks.

**Strengths:**

1.  The paper is grounded in a clear and compelling empirical analysis (Figure 1). The two observations (non-uniqueness and heterogeneity) are well-demonstrated and provide a strong "why" for the proposed method.
2.   The proposed solutions map directly to the identified problems. The adapter-level supervision (direction and spectral loss) is a very clever way to address the non-uniqueness issue. The authors also correctly note the importance of an *efficient* implementation (Appendix A.3), which avoids materializing the $d \times d$ matrix and makes the approach practical.
3.   The primary goal of such a model is to generalize to new tasks. The 5% absolute improvement on zero-shot generation (Table 3) over the T2L baseline is a significant and meaningful result, demonstrating the value of the structure-aware approach.
4.  The method is shown to be effective on both encoder-decoder (FLAN-T5) and decoder-only (Gemma-2) architectures, suggesting the principles are general.
5.  The paper is well-organized, and the progression from observation to method to results is logical and easy to follow.

**Weaknesses:**

1. The main weakness is the ablation study in Table 4. The model with just the MoE decoder and a standard reconstruction loss ("X", "X", "✓") achieves 95.2% average accuracy. The full model, with the novel adapter-level losses ("✓", "✓", "✓"), achieves 96.0%. This 0.8% difference on in-distribution tasks seems to *undermine* the importance of the adapter-level supervision ($L_{ang}$ and $L_{spec}$), which is presented as a primary contribution.
2.  Related to Weakness #1, the paper argues that the adapter-level losses are critical for *generalization* and avoiding "memorizing" specific decompositions, which is key for zero-shot performance. However, the ablation study in Table 4 is *only* performed on in-distribution tasks. The most critical ablation—showing the zero-shot performance of the "MoE + reconstruction" model (the 95.2% one)—is missing. Without this, the central claim that the novel losses are *necessary* for zero-shot generalization is not fully substantiated.
3.  The "Average LoRA" baseline results in Table 2 are clearly an error. The values appear to be copy-pasted from Table 1, and they show scores (e.g., 96.8 on ArcC) that are vastly higher than the "Task-specific LoRAs" (the oracle, 76.7). This is a sloppy error that should have been caught.

**Questions:**

1.   The ablation in Table 4 suggests your novel adapter-level losses ($L_{ang}$, $L_{spec}$) provide only a marginal (0.8%) benefit on in-distribution tasks over a standard reconstruction loss when the MoE decoder is used. You motivate these losses as being essential for zero-shot generalization. To support this claim, please provide the **zero-shot ablation results** for the seven unseen tasks (the same setup as Table 3). Specifically, what is the zero-shot performance of the model with *only* the MoE decoder and a reconstruction loss (the one that scored 95.2% in Table 4)? This is crucial to validate your central contribution.
2.  Please correct the "Average LoRA" baseline results in Table 2. The current values are nonsensical, as they significantly outperform the task-specific oracle.
3.  Have you analyzed the expert specialization in your MoE decoder? The routing is based on module and layer embeddings, motivated by Obs-2 (heterogeneity). Do the experts indeed learn to specialize on specific module types (e.g., encoder vs. decoder) or spectral-entropy profiles as hypothesized?
4.  What is the training time and parameter-count overhead of LoRAGen (for Stage 1) compared to the T2L baseline? How much do the adapter-level losses and MoE decoder add to the computational cost?

---

> ### Author Response · Authors · 2025-11-25
> **Response to Reviewer miEg (part 1)**
>
> Thanks for your feedback. We have summarized your main concerns into four questions and revised the paper accordingly. We hope the following responses can address your concerns.
>
> ---
>
> **Q1: Are the adapter-level losses $L_{\text{ang}}$ and $L_{\text{spec}}$ really important, especially for zero-shot generalization?**
>
> > **Weakness 1.** The model with just the MoE decoder and a standard reconstruction loss (“X”, “X”, “✓”) achieves 95.2% average accuracy, while the full model with adapter-level losses (“✓”, “✓”, “✓”) achieves 96.0%. This small 0.8% improvement on in-distribution tasks appears to undermine the importance of $L_{\text{ang}}$ and $L_{\text{spec}}$.
> > **Weakness 2 / Question 1.** Since the paper claims that these losses are key for zero-shot generalization, please report the zero-shot performance of the “MoE + reconstruction only” model on the seven unseen FLAN tasks (Table 3 setup).
>
> **Response:** We agree that the crucial test is zero-shot performance on unseen tasks. In the revised paper, we add a **zero-shot ablation** on the same seven held-out FLAN tasks as in Table 3. The results are as follows:
>
> | $L_{\text{ang}}$ | $L_{\text{spec}}$ | $D_\theta$ | AP-Rec | AP-Pos | QASC-1 | QASC-2 | WQA-T | WQA-A | Avg. (acc) |
> |:---------------:|:-----------------:|:----------:|:------:|:------:|:------:|:------:|:-----:|:-----:|:----------:|
> | ✗ | ✗ | ✓ | 72.7 | 35.9 | 24.5 | 10.4 | 10.2 | 65.9 | 36.6 |
> | ✓ | ✓ | ✓ | **75.1** | **42.2** | **28.1** | **14.5** | **14.3** | **67.2** | **40.2** |
>
> The average zero-shot accuracy improves from **36.6%** to **40.2%**, a **+3.6** absolute gain (around **+10% relative**). The gains are especially large on the more challenging reasoning tasks: **QASC-1**: 24.5 → 28.1 (+3.6), **QASC-2**: 10.4 → 14.5 (+4.1), **WQA-T**: 10.2 → 14.3 (+4.1)
>
> These results also clarify our earlier observation:
>
> - On **in-distribution** tasks, the MoE decoder already captures most structural information, so adding $L_{\text{ang}}$ and $L_{\text{spec}}$ yields a modest +0.8% refinement.
> - On **unseen** tasks, directly supervising the full adapter $\Delta W$ with direction and spectral losses prevents the model from relying on arbitrary low-rank decompositions and significantly improves generalization.
>
> We now report this ablation in the Appendix A.2 (Table 5) and highlight its implication in Section 5.3.
>
> ---
>
> **Q2: The “Average LoRA” baseline in Table 2 is clearly incorrect.**
>
> > **Weakness 3 / Question 2.** The “Average LoRA” results in Table 2 appear to be copied from Table 1 and even outperform the task-specific oracle, which is nonsensical. Please correct this baseline.
>
> **Response:** We thank you for pointing out this mistake. The “Average LoRA” row in the original Table 2 was indeed accidentally copied from Table 1. We have recomputed all metrics and corrected the table.
>
> In the **revised Table 2**, the Average LoRA baseline achieves an average accuracy of **70.0%**, which is:
>
> - lower than the task-specific LoRAs (oracle, **74.5%**),
> - lower than our method LoRAGen (**72.7%**),
> - but higher than the base Gemma-2-2B-Instruct model and other baselines (D2NWG, T2L).
>
> Therefore, after correction, the main conclusions of Table 2 remain unchanged: LoRAGen consistently outperforms Average LoRA, D2NWG, and T2L, and is competitive with task-specific LoRAs.

---

> ### Author Response · Authors · 2025-11-25
> **Response to Reviewer miEg (part 2)**
>
> **Q3: Do the MoE experts actually specialize according to module type / spectral profile as motivated by Obs-2?**
>
> > **Question 3.** Since the MoE routing uses module and layer embeddings and is motivated by heterogeneous spectral properties, it is important to check whether experts actually specialize to particular module types (encoder, decoder, cross-attention) or spectral-entropy profiles.
>
> **Response:** We add a new **Weight Distribution Analysis** for both base models in the revised Appendix A.3. For FLAN-T5-Large, we analyze the LoRAs corresponding to the seven FLAN tasks used in Table 1. For Gemma-2-2B-Instruct, we analyze the LoRAs corresponding to the eight benchmark tasks used in Table 2. In each case we compare three types of adapters: (1) task-specific (oracle) LoRAs, (2) LoRAGen (full model), and (3) LoRAGen without the MoE decoder (a single shared decoder).
>
> **(a) Spectral entropy distribution analysis.**
>
> For each adapter $\Delta W_{m,\ell}$, we compute its spectral entropy $H_{\text{spec}}(\Delta W_{m,\ell})$.
>
> - **FLAN-T5-Large (encoder–decoder).**
>   We group adapters by module type (Encoder, Decoder-Self, Cross-Attention) and plot the entropy distributions per module.
>   - For **task-specific LoRAs**, we observe clear heterogeneity: encoder modules have the highest spectral entropy, decoder self-attention the lowest, and cross-attention lies in between.
>   - For **LoRAGen (full)**, the generated adapters **preserve the same pattern** of spectral entropy across modules, indicating that different experts in the MoE decoder learn module-specific spectral profiles.
>   - For **LoRAGen without MoE**, the three module types collapse into **almost identical high-entropy distributions**, showing that a single shared decoder fails to capture the heterogeneous weight distributions and instead learns a nearly uniform pattern.
>
> - **Gemma-2-2B-Instruct (decoder-only).**
>   Here there is a single module type (decoder blocks), so we plot the spectral entropy distributions for decoder adapters for the three adapter sets listed above.
>   - Task-specific LoRAs concentrate around relatively high entropy values.
>   - LoRAGen (full model) shows a very similar profile with a slightly higher-entropy peak.
>   - In contrast, removing the MoE decoder shows a lower entropy and yields a much broader distribution, suggesting that a single shared decoder fails to match the spectral distribution of task-specific LoRAs.
>
>
> These spectral distributions are visualized in the revised **Appendix A.3 (Figure 4 and Figure 5).**
>
> **(b) Layer-wise cosine similarity analysis.**
>
> We further examine how closely the generated adapters align with the task-specific ones at each layer. For each layer index and module type, we compute the cosine similarity between the generated adapter and the corresponding task-specific adapter, averaged over the same tasks as in Tables 1 and 2.
>
> - With the LoRAGen (full model), cosine similarities are **consistently positive** for FLAN-T5-Large and Gemma-2-2B. This indicates that different experts specialize not only by module type (for FLAN) but also by layer, capturing fine-grained adaptation patterns.
>
> - Without MoE decoder, the cosine similarities are **close to zero across nearly all layers**, for both base models, meaning that the shared decoder does not recover the detailed layer-wise structure, even though it can still obtain reasonable in-distribution accuracy.
>
> These new visualizations are included in the revised **Appendix A.3 (Figure 6 and Figure 7).** Totally, the spectral-entropy and cosine-similarity analyses confirm that the MoE decoder indeed learns experts specialized in module type and spectral profile, as hypothesized from Obs-2 (heterogeneity).

---

> ### Author Response · Authors · 2025-11-25
> **Response to Reviewer miEg (part 3)**
>
> **Q4: What are the training-time, parameter, and computational overheads of LoRAGen compared to T2L?**
>
> > **Question 4.** Please report the training time and parameter-count overhead of LoRAGen (Stage 1) relative to T2L, and clarify how much extra computational cost the adapter-level losses and MoE decoder introduce.
>
> **Response:** We have added a detailed efficiency discussion in Appendix A.5.
>
> 1. **Practical training time (single NVIDIA A40, 40GB).**
>
> - For the **in-distribution experiments** in Tables 1 and 2, the stage-1 training times for T2L and LoRAGen are about 2 and 2.5 hours, respectively.
>
> - For the **zero-shot experiments** in Table 3, the total training time of both T2L and LoRAGen is **less than one day** on the same A40 GPU.
>
> 2. **Parameter counts of the generator.**
>
> - With FLAN-T5-Large as the base model:
>   - T2L parameters: about 50M;
>   - LoRAGen parameters: about 53M in total (15M activated parameters).
>
> - With Gemma-2-2B as the base model:
>   - T2L parameters: about 30M;
>   - LoRAGen parameters: about 69M in total (22M activated parameters).
>
>
>
> For **FLAN-T5-Large**, T2L and LoRAGen have comparable generator sizes (50M vs. 53M), while LoRAGen already achieves better in-distribution and zero-shot performance. For **Gemma-2-2B-Instruct**, we intentionally use a larger generator for LoRAGen (69M) than for T2L (30M) so that the MoE decoder has enough capacity to generate all four types LoRA parameters, namely qa, qb, va, and vb.
>
> 3. **Memory usage.**
>
> During training on an NVIDIA A40 GPU:
>
> - Enabling the **MoE decoder** introduces about **1000 MiB** of additional GPU memory usage.
> - Adding the **adapter-level losses** introduces about **300 MiB** of additional GPU memory usage.
> - Taking these components together, LoRAGen training in practice requires around **2500 MiB** of GPU memory.
>
> 4. **Theoretical complexity.**
>
> We clarify in Appendix A.4 that adapter-level supervision is implemented *without* producing the full $d \times d$ matrix $\Delta W = BA^\top$:
>
> - Reconstruction on low-rank decomposition matrices $A, B \in \mathbb{R}^{d \times r}$ costs $O(dr)$.
> - The **direction loss** (cosine similarity on $\Delta W$) is computed as a quadratic form with complexity $O(dr^{2})$, avoiding $O(d^{2})$ cost.
> - The **spectral loss** uses a reduced QR decomposition and an SVD on an $r \times r$ matrix, giving complexity $O(dr^{2} + r^{3})$.
>
>
> Since $r \ll d$ in our setting, these adapter-level losses increase the per-layer cost by at most a factor of $O(r)$ compared with standard reconstruction on $A,B$, while still avoiding any $O(d^{2})$ operations, which matches the acceptable training-time increase observed in practice.

---

### Author Response · Authors · 2025-12-03
**General Comments by Authors**

We sincerely appreciate the insightful feedback provided by all reviewers, which has helped us revise the manuscript. Below, we highlight the **key novelty** of our approach, the contributions of **LoRAGen**, and our responses to the reviewer’s concerns.

***1. Key Novelty***

**LoRAGen** is the **first weight-space learning method specifically tailored to LoRA generation**, combining **full adaptation matrix supervision** with a **module-aware MoE decoder** to achieve strong **in-distribution performance** and **non-trivial zero-shot gains** across both **encoder-decoder** and **decoder-only backbones**.

Specifically, our design introduces **supervision on the full adaptation matrix $\Delta W$**, which addresses the non-uniqueness of low-rank decomposition matrices, resulting in **improved generalization**, especially for unseen tasks in **zero-shot settings**. The **module-aware MoE decoder** specializes in different module types (e.g., encoder, decoder self-attention, cross-attention) by utilizing module and layer embeddings, capturing the **heterogeneous distributions** of LoRA weights.

> **Reviewer miEg**: "The adapter-level supervision (direction and spectral loss) is **a very clever way** to address the non-uniqueness issue. The **5% absolute improvement** on zero-shot generation (Table 3) over the T2L baseline is **a significant and meaningful result**, demonstrating the value of the structure-aware approach."

> **Reviewer 6LiF**: "The empirical results are **promising.**"

> **Reviewer mT9U**: "This approach allows different experts to specialize in generating weights for distinct components, which is a **novel and highly effective concept**. Furthermore, the paper introduces a **unique adapter-level supervision strategy** (using direction and spectral losses) to directly address the ambiguity of low-rank matrix factorization—**a subtle but critical problem** that previous methods have largely overlooked."

***2. Contribution to LoRA Parameter Generation***

LoRAGen makes a significant contribution to the field by introducing a novel **generation framework** that improves upon previous weight generation methods. The combination of the module-aware MoE decoder and adapter-level supervision provides a more **principled way** to learn the **geometry of the LoRA weight space**, allowing for more **generalizable** and **scalable LoRA generation**. This design is grounded in solid empirical analysis, specifically focusing on the **non-uniqueness of low-rank decompositions** and **heterogeneous weight distributions** across different network modules.

> **Reviewer mT9U**: "A **Novel and Robust Generation Framework**: The paper significantly improves upon previous weight generation methods by introducing a more robust and structure-aware framework."

> **Reviewer miEg**: "The method is shown to be effective on both encoder-decoder (FLAN-T5) and decoder-only (Gemma-2) architectures, suggesting **the principles are general.**"

***3. Responses to Reviewer nPHN’s Concerns***

We have carefully addressed **all of** Reviewer nPHN's feedback, particularly around the **clarification of the non-uniqueness in $\Delta W$**. As requested, we present **new results** showing that **supervision on $\Delta W$** significantly improves **generalization to unseen tasks** in **zero-shot settings**.

- Our revised **zero-shot ablation study** (Table 5) demonstrates that **supervising on $\Delta W$** improves **zero-shot accuracy** by **+3.6%** (from **36.6%** to **40.2%**). This result provides strong evidence that **adapter-level supervision** is crucial for achieving **non-trivial gains** in **zero-shot performance**.

> Reviewer **miEg**: *“The **5% absolute improvement on zero-shot generation** (Table 3) over the T2L baseline is **a significant and meaningful result**, demonstrating the value of the structure-aware approach.”*

>  Reviewer **6LiF**: *“The empirical results are **promising**.”*

>  Reviewer **mT9U**: *“Crucially, it demonstrates **excellent generalization** in both in-distribution and **challenging zero-shot** (out-of-distribution) settings, proving that it learns a meaningful mapping from language to weights rather than just memorizing trained adapters.”*

---

### Meta-Review · Area_Chair_LR3n · 2026-01-03

**Summary:**

This paper presents a model to generate LoRA parameters for text-described downstream tasks, via a specialize hypernetwork. This hypernetwork is framed as a conditional "diffusion" model where one main contribution lies in the design of the training loss: instead of applying supervision on LoRA's A and B matrices separately, this work relaxes the objective to let the hypernetwork output mimic the product $A\cdot B$. Such a design is motivated by the fact that many distinct pairs of (A, B) can produce identical product $A\cdot B$ (e.g., the product is invariant up to rotation, etc). An MoE module is also integrated to make the proposed LoRAGen generate parameters for heterogeneous base modules.

The authors have provided comprehensive response to all reviewers. Especially, Reviewer nPHN who initially gave a "reject" acknowledged the resolution of the main concerns (which I agree). Overall, the main concerns by all reviewers have been addressed reasonably well.

After reading the paper carefully, I agree that this paper presents valuable contribution and indeed improves prior arts (T2L) in the same problem domain. Yet some technical details remain unclear (additional questions not mentioned by the reviewers):

* Although Fig 1 suggests a stronger correlation between $\Delta W$ and task similarity than $A$ and $B$ alone, it also reveals a fundamental question on whether the loss terms really captures the LoRA weight structure. The y-axis ("LoRA cosine similarity") for the $\Delta W$ plot has very low values (close to 0) -- almost indicating the similarity between random matrices. This shows that
  * the simple cosine similarity does not capture the structure of LoRA's $\Delta W$, casting a doubt on whether Fig 1 is really a valid motivation for the proposed design
  * More importantly, if cosine similarity is not a good metric, why the proposed "Direction loss" & "Spectral loss" are sufficient measures for the weight structure? e.g., "Direction loss" is also just based on cosine similarity.

* "Diffusion" formulation: It is unclear how the proposed design is cast under the "diffusion" framework. There lacks a description on the encoding process, and the loss in Eq 3 and 4 are not ELBO bounds. Specifically, I wonder if removing the "Encoder" module in Figure 2 as a whole can also work. It seems we can design a "normal" hypernetwork by generating LoRA parameters directly from the "task description" embeddings. The noise sampling & the "diffusion" framing do not necessarily seem like playing an important role given the description in Sec 4.

Overall, I agree with the reviewers that this paper makes some meaningful contribution to the community, and hence recommend an "accept". However, it would be more convincing if the above question are also addressed.

**Reviewer Concerns:**

# Reviewer miEg

Many of the concerns have been addressed. Yet I remain not full convinced about the impact of adapter-level losses $L_\text{ang}$ and $L_\text{spec}$. The authors have argued that the performance gains brought by those structural losses are more significant on unseen tasks. This is true, but it still doesn't explain why for the in-domain case, the training without $L_\text{ang}$ and $L_\text{spec}$ can achieve such a high accuracy (95.2 on avg). From Eq 3 and 4, without $L_\text{ang}$ and $L_\text{spec}$, the training does not have *any* supervision from the good / ground-truth LoRA parameters, meaning that the generated parameters should be of very low quality. It seems very unlikely that such a model would learn anything meaningful to produce the 95.2 accuracy.

# Reviewer nPHN

The reviewer has acknowledged the resolution of most of the initial concerns, and I agree with such judgement. The reviewer mentioned 2 remaining concerns, which I also find reasonable:
* $\Delta W$ is indeed non-unique, even through training on $\Delta W$ would be more stable than on $A$ and $B$ separately.
* The practical value of this approach is not immediately clear, especially given the performance gap on unseen tasks. However, I think it also makes sense to leave it as future work to close such a gap, and leave it as an open question for now.


# Reviewer 6LiF

Most concerns have been addressed (as have been acknowledged by the reviewer themselves).


# Reviewer mT9U

The authors have addressed many of the concerns. The reviewer asked about comparison on other generative frameworks, and I remain curious on the real benefits from the authors' diffusion formulation. As mentioned in the "Summary", it is not clear how to interpret the proposed generative process (diffusion? VAE? or something mixed?), and thus the reviewer's concern on the comparison has not been fully addressed.

**Reviewer Scores:**

I think the reviewers miEg, 6LiF and mT9U will maintain their scores, while nPHN may slightly increase the initial rating.

---

### Decision · Program_Chairs · 2026-01-26

Accept (Poster)